# MicroSEC filters sequence errors for formalin-fixed and paraffin-embedded samples

Masachika Ikegami [1,2,3✉], Shinji Kohsaka [1✉], Takeshi Hirose [1,4], Toshihide Ueno [1], Satoshi Inoue[1], Naoki Kanomata[5], Hideko Yamauchi[6], Taisuke Mori[7], Shigeki Sekine[7], Yoshihiro Inamoto[8], Yasushi Yatabe [7,9], Hiroshi Kobayashi[2], Sakae Tanaka [2] & Hiroyuki Mano [1✉]

The clinical sequencing of tumors is usually performed on formalin-fixed, paraffin-embedded samples and results in many sequencing errors. We identified that most of these errors are detected in chimeric reads caused by single-strand DNA molecules with microhomology. During the end-repair step of library preparation, mutations are introduced by the mis-annealing of two single-strand DNA molecules comprising homologous sequences. The mutated bases are distributed unevenly near the ends in the individual reads. Our filtering pipeline, MicroSEC, focuses on the uneven distribution of mutations in each read and removes the sequencing errors in formalin-fixed, paraffin-embedded samples without over-eliminating the mutations detected also in fresh frozen samples. Amplicon-based sequencing using 97 mutations confirmed that the sensitivity and specificity of MicroSEC were 97% (95% confidence interval: 82–100%) and 96% (95% confidence interval: 88–99%), respectively. Our pipeline will increase the reliability of the clinical sequencing and advance the cancer research using formalin-fixed, paraffin-embedded samples.

[1] Division of Cellular Signaling, National Cancer Center Research Institute, Tokyo, Japan. [2] Department of Orthopaedic Surgery, Faculty of Medicine, The University of Tokyo, Tokyo, Japan. [3] Department of Musculoskeletal Oncology, Tokyo Metropolitan Cancer and Infectious Diseases Center Komagome Hospital, Tokyo, Japan. [4] Department of Orthopaedic Surgery, Graduate School of Medical Sciences, Kyushu University, Fukuoka, Japan. [5] Department of Pathology, St Luke's International Hospital, Tokyo, Japan. [6] Department of Breast Surgical Oncology, St Luke's International Hospital, Tokyo, Japan. [7] Division of Molecular Pathology, National Cancer Center Research Institute, Tokyo, Japan. [8] Department of Hematopoietic Stem Cell Transplantation, National Cancer Center Hospital, Tokyo, Japan. [9] Department of Biobank and Tissue Resources, National Cancer Center Research Institute, Tokyo, Japan. ✉email: ikegami-tky@umin.ac.jp; skohsaka@ncc.go.jp; hmano@ncc.go.jp

Cancer gene panel testing using next-generation sequencing has been applied in routine practice to identify the somatic as well as germline mutations and to determine the appropriate treatment strategy for cancer patients[1]. Somatic mutations often only occur in a small subset of cells and are present in a small fraction of DNA molecules from tumor samples. Somatic mutations can be detected in fresh or fresh frozen (FF) samples using next-generation sequencers with low error rate, but those materials are not always available for clinical sequencing.

As a result, the nucleic acids extracted from formalin-fixed and paraffin-embedded (FFPE) tumor tissues collected in surgeries or biopsies are more commonly used[2–4]. Formalin fixation and prolonged storage cause various changes in nucleic acids, such as the cross-linking between nucleic acids and proteins, denaturation, cytosine deamination, and chemical modification[5]. As a result, the DNA extracted from FFPE tissues is usually fragmented and contains single-stranded DNA (ssDNA)[6]. Low DNA quality causes substantial noise in the sequencing reaction. Therefore, it is highly challenging to detect the mutations that occur at low variant allele fractions (VAFs) in FFPE samples[7–9]. However, the development of filtering pipelines for FFPE artifacts has not progressed remarkably beyond the well-known CG-to-TG mutation caused by cytosine deamination[10–14].

Here we propose a major mechanism for FFPE sequencing errors, microhomology-induced chimeric read (MICR) formation in capture-based target sequencing. MICRs are ssDNA-derived artifacts and classified into two types induced by different mechanisms (Fig. 1). The first type of MICRs results from the hairpin structure formed by two palindromic sequences in the same ssDNA molecule. The second type of MICRs is formed from the mis-annealing of two ssDNA molecules derived from different homologous regions. MICRs are formed during the end-repair step of library preparation for clinical sequencing, wherein a considerable amount of extracted DNA is denatured to ssDNA and behaves as site-directed mutagenesis polymerase chain reaction (PCR) primers[15]. Based on our theory that artifacts are derived from ssDNA-annealing, we have developed a MICR-originating Sequence Error Cleaning pipeline (MicroSEC), a post hoc filtering pipeline to predict whether a given mutation is an MICR-derived error. This pipeline allows the processing of thousands of mutations of target sequencing data within hours on a standard PC with 16 gigabytes of memory. MicroSEC requires a list of mutations and corresponding BAM files, rather than FASTQ files as it uses the positional bias of reads mapped against mutations.

## Results and discussion
### Examples of artifacts and the presumed mechanisms.
To better understand the spectrum of FFPE artifacts, we performed target sequencing of a low-quality FFPE sample of normal breast tissue using a 478-gene panel and reviewed likely artifacts. First, we found mapping anomalies characteristic of artifacts in FFPE samples. DNA extracted from samples was fragmented at random positions to the appropriate size before sequencing. Mutated bases are expected to be distributed evenly in the reads. However, we observed a T-to-C artifact in *FGFR4* gene with a marked bias in the position of the mutation (Fig. 2a). In the case of all reads with the artifact, only six bases downstream of the mutation were mapped, and the rest were soft-clipped. This phenomenon was not observed in the non-mutated reads. The mapping of a representative read with an artifact in *FGFR4* was examined in detail (Fig. 2b). The upstream sequence of the read was mapped to the forward strand of the genome, and the downstream sequence was mapped to the reverse strand of the same genomic

region. Strangely, the upstream and downstream sequences overlapped, as did the genomic sequences to which each was mapped. Two palindromic sequences exist in close proximity to each other in this region. From this, we estimated the phenomena shown in Fig. 2c in the end-preparation step of library preparation. A ssDNA containing two palindromic sequences potentially formed a hairpin structure. After nicking and partial denaturation, the double-stranded DNA could be regenerated by DNA polymerase. Then, the mismatched base between two palindromic sequences was detected as a mutation.

The following phenomena are observed when reviewing all the reads supporting a mutation. In the case of a true mutation, the altered bases are expected to be evenly distributed across the reads. Conversely, in case of an MICR error, the altered bases are distributed preferentially near the effective ends of the read, with severely limited lengths that match the reference sequence upstream or downstream of the mutations. The VarScan2 mutation caller empirically employs whether the mutated bases are biased toward the ends of the reads as a filtering metric, but it does not take soft-clipping into account[16]. However, the details of such a bias is vaguely known. As MICRs form before the adapter ligation step in the library preparation, it cannot be ruled out in principle using template tagmentation techniques such as molecular identifier (ID) barcoding[1]. Therefore, there is a strong need for specifically designed filtering pipelines.

### Study design.
MicroSEC is based on three criteria (Fig. 2d). First, the local palindromic sequences causing hairpin-induced errors are detected (Filter 2). Second, the distribution of mutation positions in sequencing reads is interrogated if it is too uneven to occur probabilistically (Filter 1 and 3). Third, distant regions in the genome are searched, of which sequences are exactly matched to the query FFPE reads (Filter 4).

During the step of detecting mutated base position bias, the number of bases from the mutated base to the farthest mapped base in each read is defined as the 3′ or 5′-supporting lengths (Fig. 3a). The shorter of the 3′- and 5′-supporting lengths is defined as the shorter supporting length. The probability of the mutation distribution of the supporting lengths is calculated based on the multinomial distribution; mutations with $p < 10^{-6}$ are considered artifacts (Fig. 3b). During the step of detecting hairpin-induced errors, 15-base sequences containing the mutation were extracted from each read (Fig. 3c). We considered the mutation an artifact if more than half of the sequences existed on the opposite strand within 200 bases of the neighboring sequence. During the step of detecting distant homologous regions, 40-base sequences including the mutation were extracted from each read (Fig. 3d). The mutation is considered an artifact if >15% of the sequences match another region in the genome.

We first tested the sensitivity of the algorithm using normal breast tissue, because there were few true mutations in normal mammary tissue and most mutations detected in FFPE samples were considered to be artifacts. This was followed by a testing on specificity using 23 FF and 33 FFPE breast cancer samples. After confirming the performance, MicroSEC was applied to the clinical sequencing and whole-exome sequencing data to investigate the usefulness of MicroSEC in actual clinical practice. Since MICR-originating artifacts were not produced by PCR, amplicon-based sequencing was used as an external validation of MicroSEC.

We examined the performance of MicroSEC in distinguishing true mutations from FFPE artifacts with our custom-made multi-gene panel test, "Todai OncoPanel"[2]. The panel including 15,600 capture probes were designed to examine 478 cancer-related genes. As the total size of target regions was 3.4 megabases, the average length of captured regions was approximately 220 base

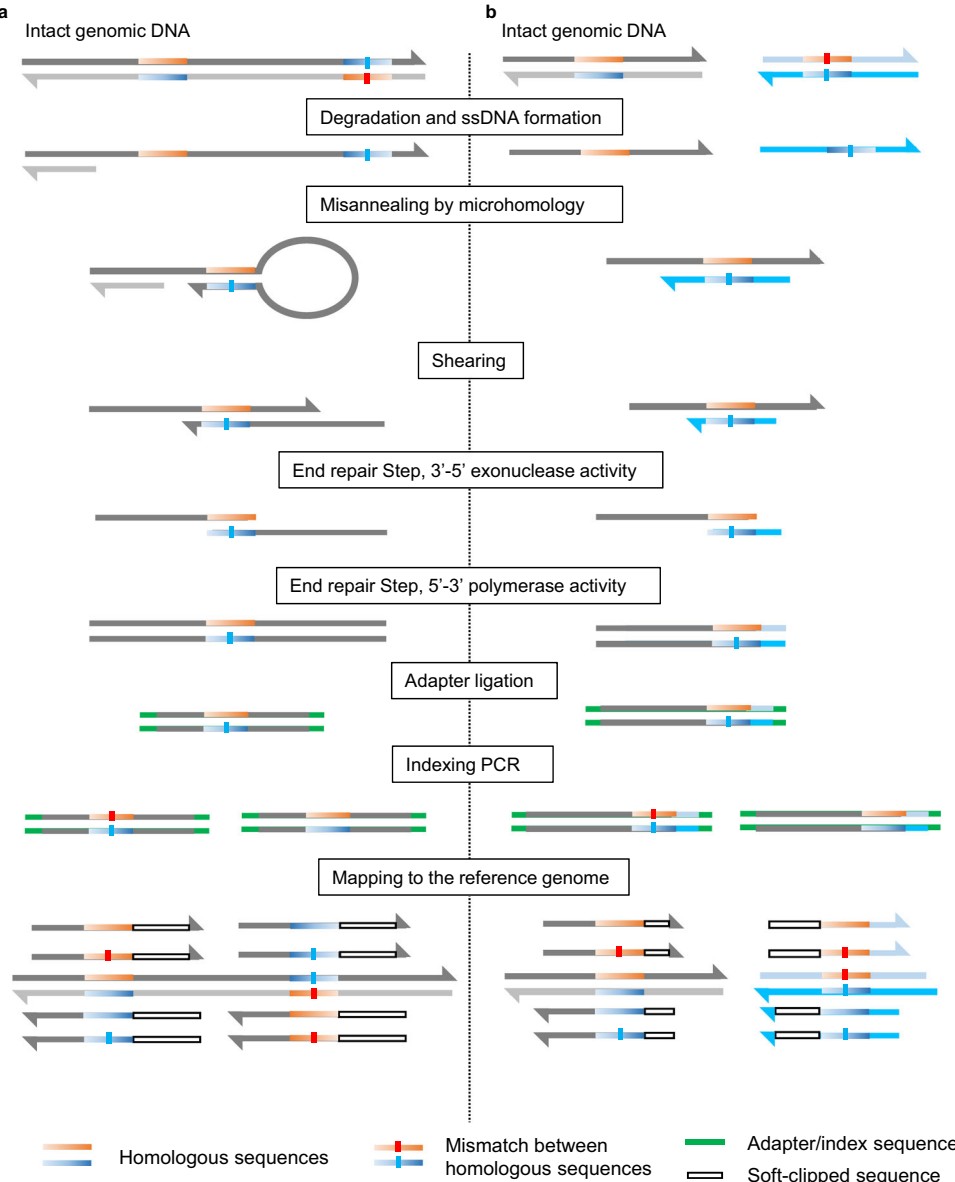

**Fig. 1 Two proposed mechanisms of microhomology-induced chimeric read (MICR) formation. a** In the region with two proximal palindromic sequences, single-stranded DNA (ssDNA) is formed after degradation and denaturation. The annealing between the two palindromic sequences forms a hairpin structure. The chimeric double-stranded DNA is formed during the shearing and end-repair steps of library preparation. Adapter and index sequences are added to the original sequence. The ligation product is amplified by polymerase chain reaction (PCR), sequenced, and mapped to the reference genome. In a chimeric read of two sequences, only one is correctly mapped, and the other is clipped by the mapping software. **b** MICR is also formed by the homologous sequences in two distant regions in the genome. Chimeric reads are generated by mis-annealing of ssDNA derived from the distant homologous regions.

pairs. We obtained FF samples of normal blood from all cases. The somatic mutations were defined as those that were identified in sample DNA but absent from matched normal blood DNA, although not all mutations detected in the blood samples were germline mutations because some mutations could be caused by clonal hematopoiesis or sequencing errors.

**Performance of MicroSEC.** To test the sensitivity of the filtering algorithm, we analyzed the target sequencing data of 53 FF and 190 FFPE normal breast tissue samples with a high mean coverage of ≥400. Our initial somatic mutation analysis pipeline identified an average of 0.3 and 11.7 somatic mutations per sample in FF and FFPE samples, respectively (Table 1). With the

MicroSEC pipeline, 0 (0%) and 10.1 (86.0%) mutations per sample were filtered out from the data of FF and FFPE samples, respectively. In FFPE samples, possible artifacts such as the CG-to-TG mutations or mutations in or adjacent to a homopolymer equal to or longer than 10-base accounted for half of the mutations that had passed through the filter (Table 1). The mutations passing through the filter had similar VAFs to the filtered out mutations (Supplementary Fig. 1a–c).

Surprisingly, two unique mutations with VAF of >50% in FFPE samples were eliminated by Filter 1 or 2 (Supplementary Table 1). Thus, a high VAF does not necessarily indicate a true mutation. The relationship between the filtering rate and mutation coverage was also examined using FFPE samples from normal breast tissues, which were expected to have few or no mutations.

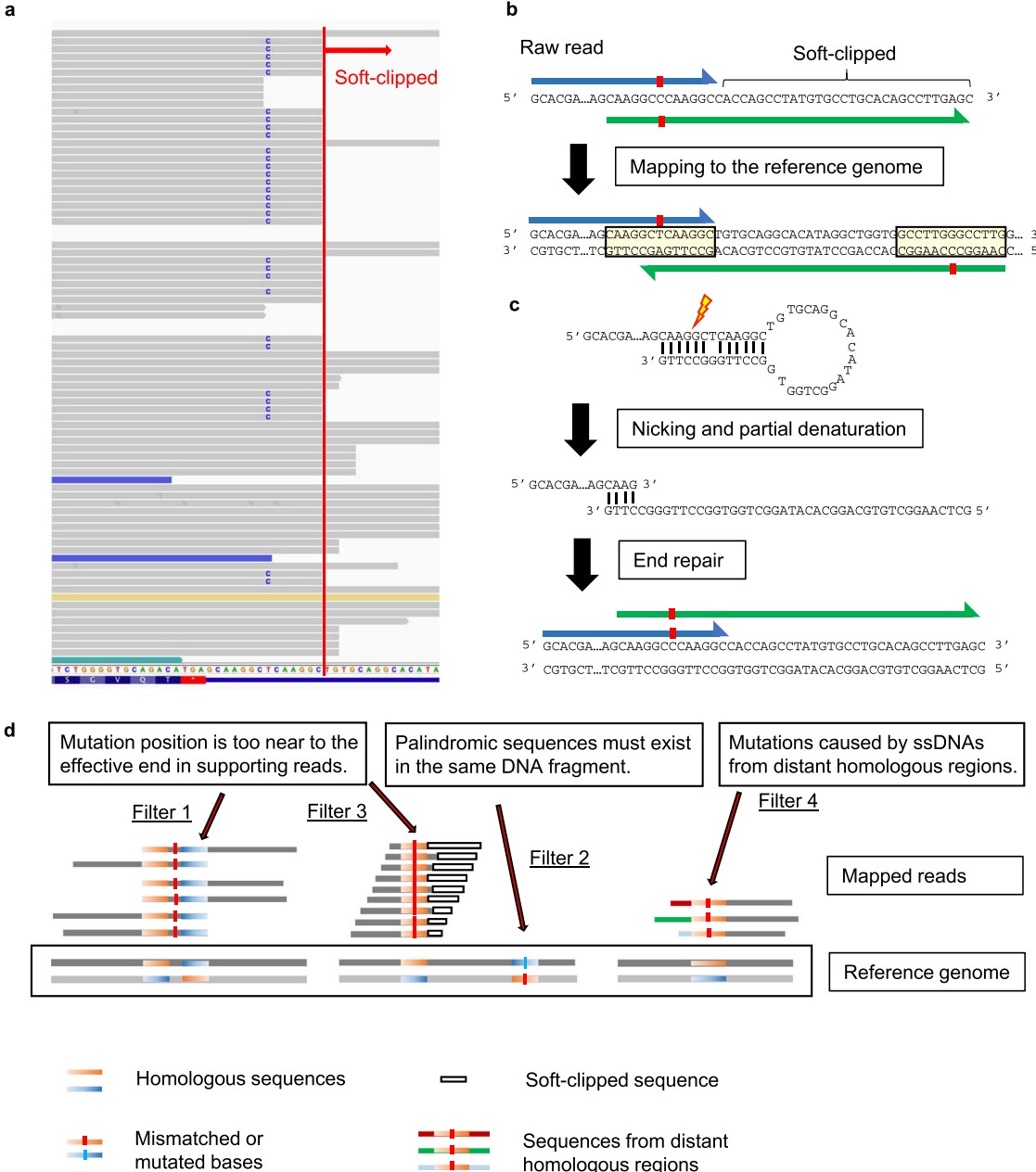

**Fig. 2 An example of microhomology-induced chimeric read (MICR)-originated sequencing error. a** The genomic sequence visualized by Integrative Genomics Viewer exhibits a T-to-C artifact in the *FGFR4* gene found in target sequencing data of a FFPE normal breast tissue sample. In all mutation-supporting reads, only six bases downstream of the mutation were mapped, and the rest is soft-clipped (red line). The blue colored read has an inferred insert size smaller than expected. The mate-reads of green or gold colored reads were mapped to different chromosomes. **b** A representative read supporting the T-to-C artifact in Fig. 2a. The upstream sequence of the read (blue arrow) was mapped to the forward strand of the genome, and the downstream sequence of the same read (green arrow) was mapped to the reverse strand. Strangely, the upstream and downstream sequences overlapped, as did the genomic sequences to which each was mapped. Since the upstream sequence was longer than the downstream sequence, only the upstream sequence was eventually mapped and the downstream sequence was soft-clipped. Two palindromic sequences exist in close proximity to each other, and the mismatched base between the two sequences (red box) represent the source of the T-to-C artifact. Most of the downstream bases were soft-clipped. **c** Presumed mechanism of the phenomenon observed in Fig. 2b. Two palindromic sequences in a single-stranded DNA (ssDNA) formed a hairpin structure at the end-repair step of library preparation. After nicking and partial denaturation, the double-stranded DNA was regenerated during the end-repair step of library preparation. The mismatched base between two palindromic sequences was defined as a mutation. **d** The MicroSEC algorithm is based on three criteria. Filter 1, 3: the distance from the mutation position to the most distant mapped base is distributed over a probabilistically improbable limited range for any reads. Filter 2: MICR-originated sequencing errors are generated when two palindromic sequences are in the same DNA fragment. Filter 4: The mis-annealing of ssDNA derived from other distant homologous regions of the genome also creates chimeric reads and artifacts. Dark-red, green, or light-blue horizontal bars represent sequences of other distant regions of the genome. Chimeric reads with mutated bases were formed.

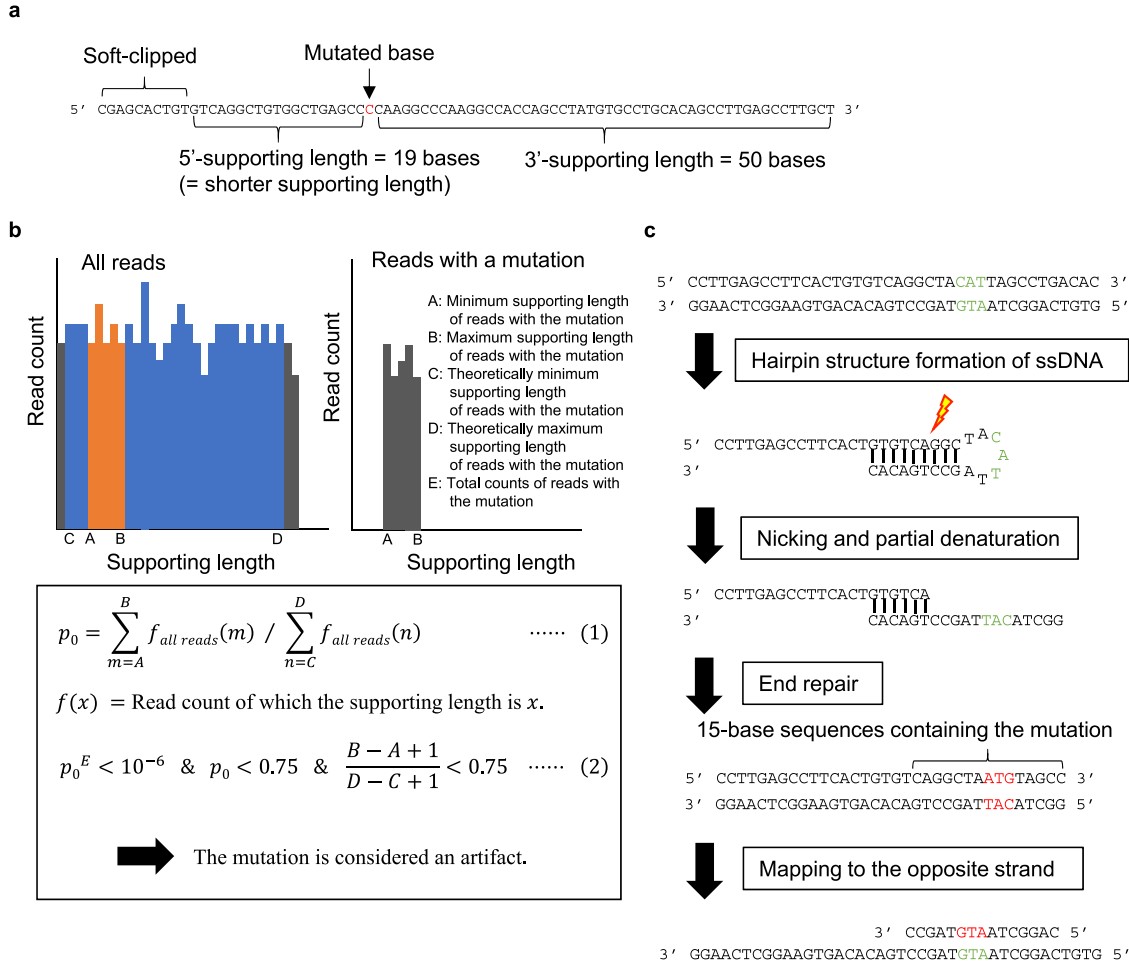

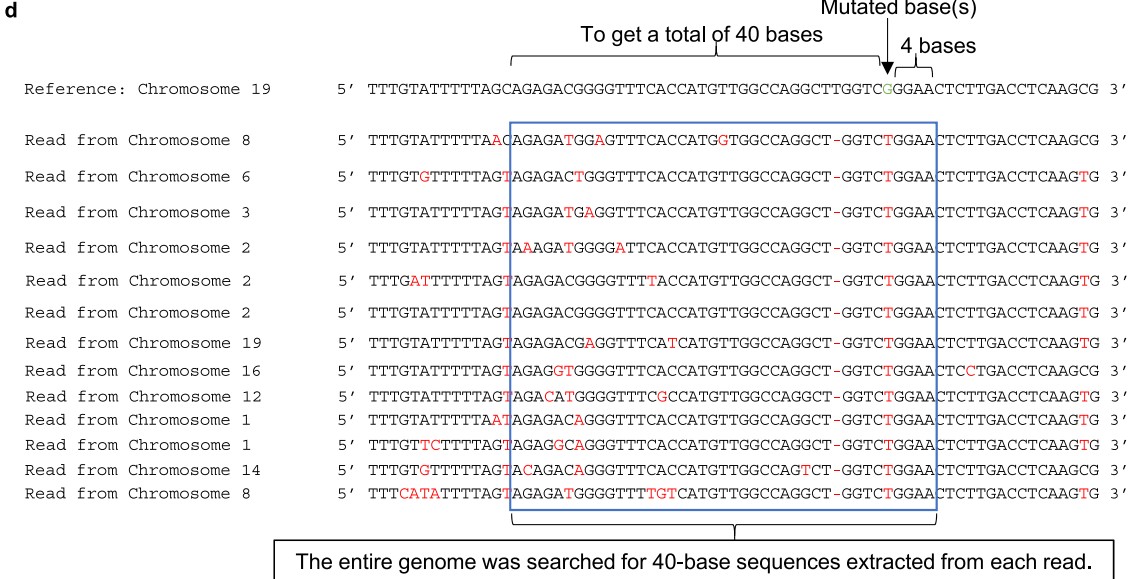

More than 90% of mutations were filtered by MicroSEC pipeline when the mutation coverage was 10–22. A tendency to decrease could be observed for the mutation filtering rate as the number of reads with mutations increased (Fig. 4).

We then tested the specificity of MicroSEC by analyzing the target sequencing data of 23 FF and 33 FFPE breast cancer samples with a high mean coverage, including 8 sets of matched FF and FFPE samples from the same patients. Our pipeline identified 4.0 and 10.7 somatic mutations per sample in the FF and FFPE samples, respectively (Table 1). All the mutations in the FF samples passed through the filter, while 3.2 mutations per sample (30.2%) were filtered out in the FFPE samples. Mutations passing through the filter tended to have a higher VAF than the filtered mutations (Supplementary Fig. 1d–f).

**Fig. 3 The details of the MicroSEC filtering criteria.** The principle of the algorithm is described with hypothetical reads. **a** Definition of supporting lengths. Supporting lengths are defined as the distances from the mutated base to the 5′ or 3′ ends of an individual read (excluding soft-clipped bases). The shorter supporting length is defined as the shorter one. **b** Filtering based on biased distribution of supporting lengths. The supporting lengths are calculated only for the reads with a mutation. A and B indicate the minimum and maximum lengths, respectively. The supporting lengths that the mapping software can theoretically generate for reads with the mutation are determined. C and D indicate the minimum and maximum values, respectively. E indicates the total count of reads with mutations. Based on the distribution of supporting lengths in reads without mutations, the probability ($p_0$) that the supporting lengths are between A and B is calculated (Eq. (1)). **c** Filtering based on suspected hairpin formation. Based on the putative mechanism of artifact generation during the end-repair step, the sequence around the mutation is derived from the opposing strand. Sequences of 15 bases containing the mutation are extracted from each read and mapped to the opposite strand within 200 bases of the neighboring sequence. If a 15-base sequence is mapped without mismatch, the read is considered to be hairpin-derived. The mutation is considered as an artifact if more than half of the reads are hairpin-derived. ssDNA single-stranded DNA. **d** Filtering for distant homologous region-derived artifacts. A G > T artifact in chromosome 19 is shown. Sequences of 40 bases containing the mutation are extracted from each read. The mutation is considered an artifact if >15% of the 40-base sequences match completely to other regions in the genome.

**Table 1 MicroSEC filtering summary for target deep sequencing.**

| | Normal breast tissue | | Breast cancer | | Clinical sequencing |
| --- | --- | --- | --- | --- | --- |
| | Fresh frozen ($N = 53$) | FFPE ($N = 190$) | Fresh frozen ($N = 23$) | FFPE ($N = 33$) | FFPE ($N = 54$) |
| *Total reads (in millions)* | 61.3 (36.6–120.4) | 83.5 (37.3–154.8) | 60.8 (43.3–114.2) | 67.5 (20.0–127.6) | 87.8 (38.0–216.0) |
| *Mapped reads (%)* | 92.2 (89.7–93.7) | 93.4 (89.4–96.8) | 91.9 (88.7–93.6) | 92.3 (89.0–94.3) | 90.5 (85.9–92.9) |
| *Unique reads (%)* | 77.1 (55.9–87.9) | 51.1 (32.6–79.8) | 76.5 (51.9–84.9) | 55.3 (38.6–86.6) | 41.1 (17.8–77.2) |
| *Mean coverage* | 928 (417–1584) | 919 (438–1544) | 944 (550–1880) | 767 (403–1525) | 817 (404–1589) |
| *Median insert size (base)* | 217 (163–312) | 158 (117–196) | 202 (158–293) | 142 (119–173) | 135 (104–177) |
| *Somatic mutations* | 0.3 (0–9) | 11.7 (0–117) | 4.0 (0–13) | 10.7 (2–34) | 21.6 (1–135) |
| Removed by | | | | | |
| Filter 1 | 0 (0–0) | 4.4 (0–88) | 0 (0–0) | 1.0 (0–9) | 0.5 (0–13) |
| Filter 2 | 0 (0–0) | 2.3 (0–30) | 0 (0–0) | 0.7 (0–4) | 0.1 (0–1) |
| Filter 3 | 0 (0–0) | 3.4 (0–73) | 0 (0–0) | 0.9 (0–6) | 0.6 (0–18) |
| Filter 4 | 0 (0–0) | 5.3 (0–40) | 0 (0–0) | 1.8 (0–9) | 3.2 (0–90) |
| Any of the Filters 1–4 | 0 (0–0) | 10.1 (0–113) | 0 (0–0) | 3.2 (0–15) | 3.6 (0–93) |
| *Mutations passing the filter* | 0.3 (0–9) | 1.6 (0–50) | 4.0 (0–13) | 7.5 (0–33) | 18.0 (0–135) |
| Filtered rate (%) | 0 | 86.0 | 0 | 30.2 | 16.6 |
| *CG-to-TG potential artifacts* | NA | 0.1 (0–12) | NA | 0.2 (0–1) | 2.6 (0–37) |
| *Intra ≥10-base homopolymer* | 0.0 (0–1) | 0.7 (0–13) | 0.0 (0–0) | 0.1 (0–2) | 0.1 (0–1) |
| Remaining mutations | 0.3 (0–9) | 0.8 (0–38) | 4.0 (0–13) | 7.1 (0–33) | 15.3 (0–128) |

Data are shown as mean (range).
NA not applicable, FFPE formalin fixed and paraffin embedded.

Notably, the MicroSEC did not remove any mutations detected in matched FF samples (Fig. 5a). Eight mutations were detected in the frozen samples only, and 42 mutations in both the frozen and FFPE samples, all of which passed through the MicroSEC filter. Sixty-five mutations were detected in FFPE samples only, of which 11 were detected as artifacts. It was necessary to elaborate that the matched samples were not collected from the exact same sites. DNA was extracted from frozen specimens of approximately 3 mm and thinly sliced FFPE specimens. The mutations found in each site were different due to tumor heterogeneity. Since frozen specimens consisted of multiple subclones, only common mutations were detected with VAF > 5%, whereas FFPE specimens comprised only a small number of subclones, subclone-specific mutations were thus also detected. The fact that mutations detected in FFPE samples had greater mutation coverage than those detected in FF samples supported this theory (Fig. 5b). This was the reason why 54 unfiltered mutations were detected in the FFPE samples, which we consider to be true mutations.

**Hyperparameter optimization.** We revealed the different distribution of the rates among low-quality bases, soft-clipped reads, artifacts from homologous regions, and mutations derived from hairpin structures in the FF and FFPE samples (Supplementary Fig. 2). We further validated the filtering hyperparameters and

thresholds. The median insert size of FFPE normal breast tissue samples for target sequencing was 158 (Table 1). We counted palindromes in FFPE normal breast samples and frozen breast tumor samples with three different ranges (150, 200, and 300 bases) to search for palindromes (Supplementary Fig. 3a). No palindromes were detected in frozen tumor samples. Using the search range of 150 bases, 430 palindromes were detected. When the search range was extended to 200 bases, we could detect 438 palindromes. Interestingly, extending the search range to 300 bases did not increase the number of detected palindromes. Based on these results, we concluded that the search range of 200 bases was appropriate. However, all analyses using any thresholds for Filter 1–4 did not filter out mutations detected in frozen tumor samples and we could not identify the optimal thresholds (Supplementary Fig. 3b–d). Further analyses with more tumor-derived mutations would thus be necessary to determine the optimal thresholds.

To evaluate the relationship between the mean coverage and artifact detection rate, we performed random sampling from the sequences of FFPE normal breast tissue samples and frozen breast cancer samples. From each library, three random sampling runs were performed so that the mean coverage of each library was 400, 300, 200, 100, 50, and 30. We calculated the average artifact detection rate after excluding mutations for which there were no reads observed. We also varied the P value threshold for Filters 1 and 3 from $10^{-2}$ to $10^{-9}$ to determine the appropriate threshold

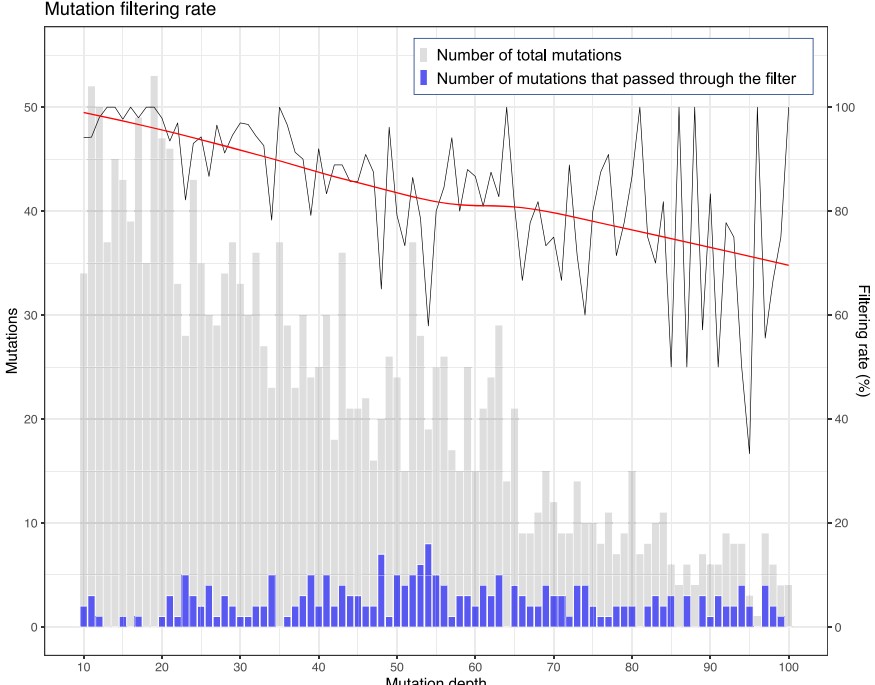

**Fig. 4 The relationship between the mutation depth and filtering efficiency in the 190 FFPE normal breast tissue samples.** The histograms of mutation depth for total mutations (gray) and mutation passing the MicroSEC filter (blue) are shown. Most of the called mutations are considered to be artifacts as normal breast tissues have little or no mutations. The mutation filtering rate (black line) is high at low depth and >90% of mutations are filtered by MicroSEC pipeline when the mutation coverage is 10–22. There is a tendency for the percentage of reads that passed the filter to increase as the number of reads with mutations increases. Local regression curve of the filtering rate is shown as a red line. The gray bars represent the number of total mutations, and the purple bars represent the number of mutations that passed through the filter. FFPE formalin fixed and paraffin embedded.

(Fig. 6). The artifact detection rate decreased, and the number of false positives increased when the average depth of coverage was <100. In addition, false positives were seen in frozen tumor samples when the $P$ value threshold was $>10^{-5}$, indicating that the threshold should be $\leq 10^{-5}$. Although the threshold could be $10^{-5}$, we adopted $10^{-6}$ to avoid overfiltering.

**Amplicon sequencing.** The MicroSEC analysis results were validated with amplicon-based sequencing that enriches target genomic regions by PCR. Ninety-seven mutations including germline mutations and low VAF ones, found in the breast tissue samples, were examined. The mutations were randomly selected from 31 FFPE normal breast tissue samples, 12 FFPE breast tumor samples, 2 FF normal breast tissue samples, and 6 FF breast tumor samples. In theory, MICR-originated artifacts cannot occur in amplification-based sequencing because PCR does not amplify small structures formed by microhomology. Consistent with this prediction, 28 out of 31 mutations that passed through the MicroSEC filter were detected by amplicon-based sequencing with a similar level of VAF, whereas 64 out of 65 filtered mutations were not detected by amplicon-based sequencing (Fig. 7 and Supplementary Data 1). With a total of 97 mutations, the sensitivity and specificity of MicroSEC in the validation study were 97% (95% confidence interval (CI): 82–100%) and 96% (95% CI: 88–99%), respectively (Fisher's exact test). Of the four mutations with discordant results, two were CG-to-TG mutations (*BRCA2* p.Arg2520* and *ESR1* p.Arg394Cys). These artifacts arose from deaminated cytosines. The Q5 DNA polymerase used in the amplicon-based sequencing could not amplify such degenerated template DNA. *NFYA* p.Gln155Pro (VAF 5.4% by target sequencing) was filtered out by MicroSEC but detected by amplicon-based sequencing at a low frequency of 1.6%. The aligned reads in capture-based sequencing visualized

by Integrative Genomics Viewer suggested that the mutation was a hairpin-derived artifact. However, it could be a true mutation (Supplementary Fig. 4a). *CENPA* p.Leu91Pro, recognized as a true mutation by MicroSEC, was not detected by amplicon-based sequencing. Of the 954 reads mapped to the mutated base in capture-based sequencing, 227 (24%) were of low quality and failed to call bases, 689 were wild type (T), 47 were C, and 1 was A (Supplementary Fig. 4b). We considered that the mutation could be an artifact caused by miscalls due to low-quality reads.

**Application of MicroSEC to clinical sequencing and whole-exome sequencing.** We further examined the utility of MicroSEC in the clinical setting by re-analyzing the sequencing data of 54 clinical FFPE samples with a high mean coverage (Table 1)[2]. At first, 21.6 somatic mutations per sample were detected by the conventional mutation caller pipeline. MicroSEC detected 3.6 artifacts per sample (16.6%), including 5 unique pathogenic mutations (Supplementary Table 2). Since annotating artifacts as pathogenic can compromise the reliability of the clinical sequencing, the post hoc filtering with MicroSEC is of great value.

We investigated the applicability of MicroSEC not only for target deep sequencing but also for whole-exome sequencing with a relatively lower coverage. Whole-exome sequencing data of 14 matched FF and FFPE samples of primary cancer (12 colorectal adenocarcinoma and 2 oral squamous cell carcinoma) was performed with a mean coverage of 199 and 255, respectively (Supplementary Table 3). Our mutation caller identified 107.0 and 118.2 mutations per sample in the FF and FFPE samples, respectively. Only 0.6 mutations per sample (0.5%) detected in the FF samples were filtered by the filter, while 10.3 mutations per sample (8.7%) were filtered out in the FFPE samples (Fig. 5c). This result suggests that the ssDNA-derived artifacts are also present in the sequencing data of frozen samples. CG-to-TG

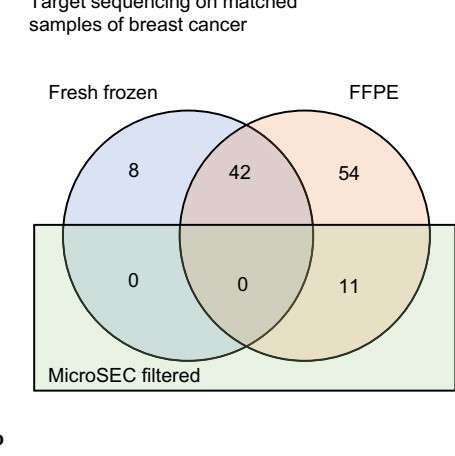

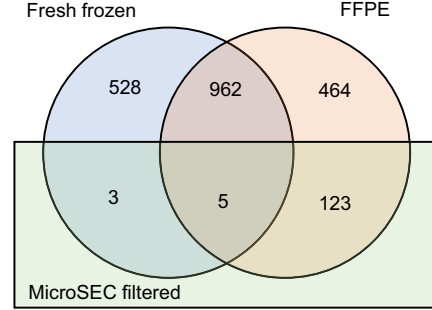

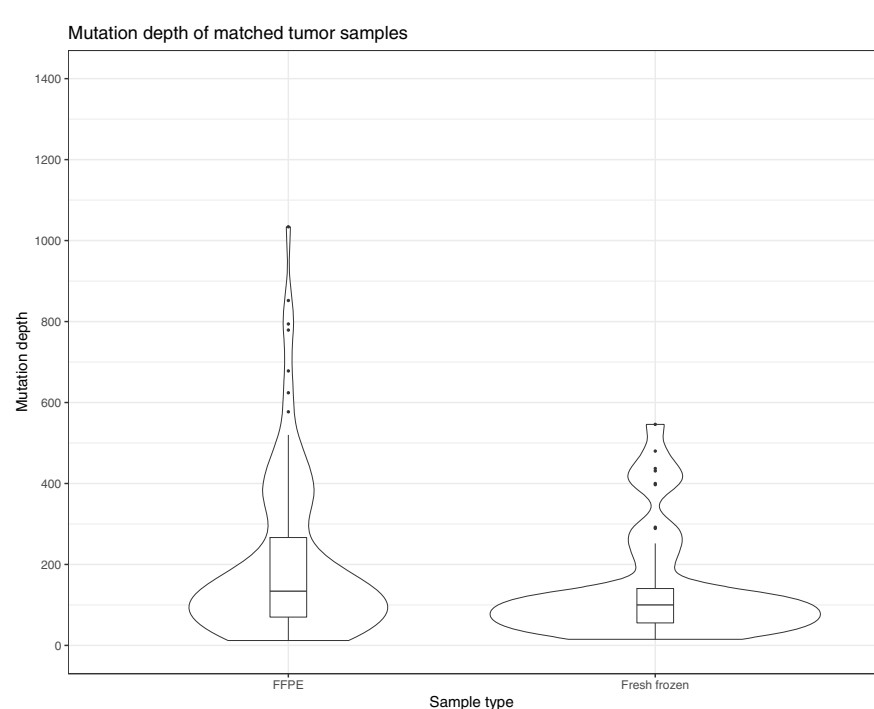

**Fig. 5 The mutations detected in matching FF and FFPE samples. a** The somatic mutations detected in the target sequencing of matching sets of frozen (blue) and FFPE (peach) breast cancer samples from eight patients. Eleven mutations found in only FFPE samples were filtered by MicroSEC (green). All mutations found in fresh frozen samples passed through the MicroSEC filter. **b** Kernel density plots and box plots of mutation depths in matched FFPE and fresh frozen breast cancer samples. The width of each kernel density plot showed the approximate frequency of the data points. In target sequencing, mutations detected in FFPE samples showed higher mutation depths than those detected in frozen samples. **c** The somatic mutations detected in the whole-exome sequencing of matching sets of frozen (blue) and FFPE (peach) primary cancer samples from 14 patients. The 123 mutations (21.0%) found in only FFPE samples were filtered by MicroSEC (green). Eight mutations (0.5%) found in fresh frozen samples were filtered by the MicroSEC filter. FFPE formalin fixed and paraffin embedded.

mutations were detected at a high rate of 38.7% of the total somatic mutations (45.8 mutations per sample), and thus most of these were possible artifacts. The ratio of CG-to-TG mutations to MicroSEC-filtered artifacts was 4.45:1, but this ratio might vary depending on the sample conditions and analysis methods. With sufficient coverage, MicroSEC was considered to be applicable for whole-exome sequencing.

## Conclusions

Most other existing error detection algorithms utilize strand bias, k-mer frequency, suffix trie, multiple sequence alignment, and statistical error models[12, 13]. The application of these algorithms is limited because they are not highly accurate. These methods disregard detailed genomic information, such as the local palindromic sequences and the position of mutations within reads. For example, Strand Orientation Bias Detector (SOBDetector)[17], the most recent filtering algorithm for FFPE sequencing artifacts, was applied to sequence data of frozen breast tumor samples and FFPE normal breast tissue samples (Fig. 8). Based on the assumption that formalin modification occurs only in one of the strands, SOBDetector detects artifacts based on the strand bias of the detected mutations, and the creators claim that it shows a state-of-the-art performance that can predict artifacts with 90% accuracy. Both SOBDetector and MicroSEC identified all 74 single-nucleotide variants (SNVs) detected in frozen breast tumor samples as true mutations. SOBDetector was able to detect only 3 of the 1351 SNVs detected in FFPE normal breast tissue

**a**

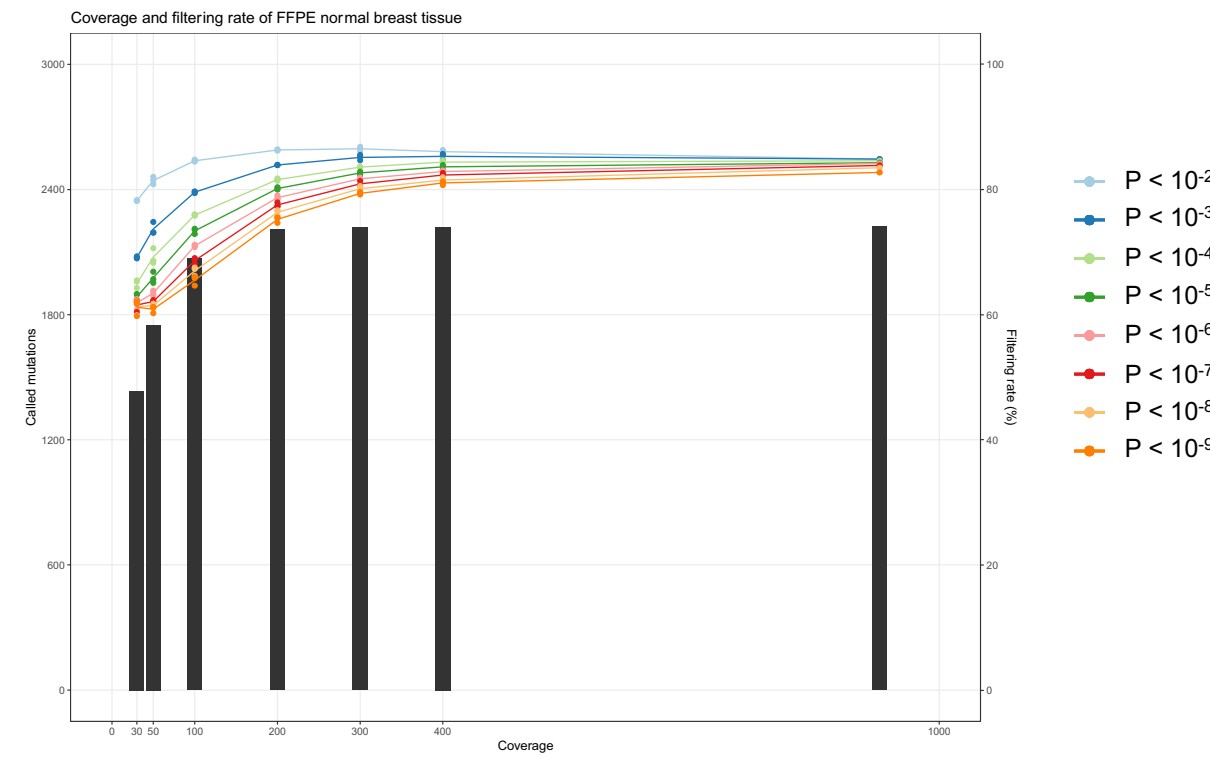

**b**

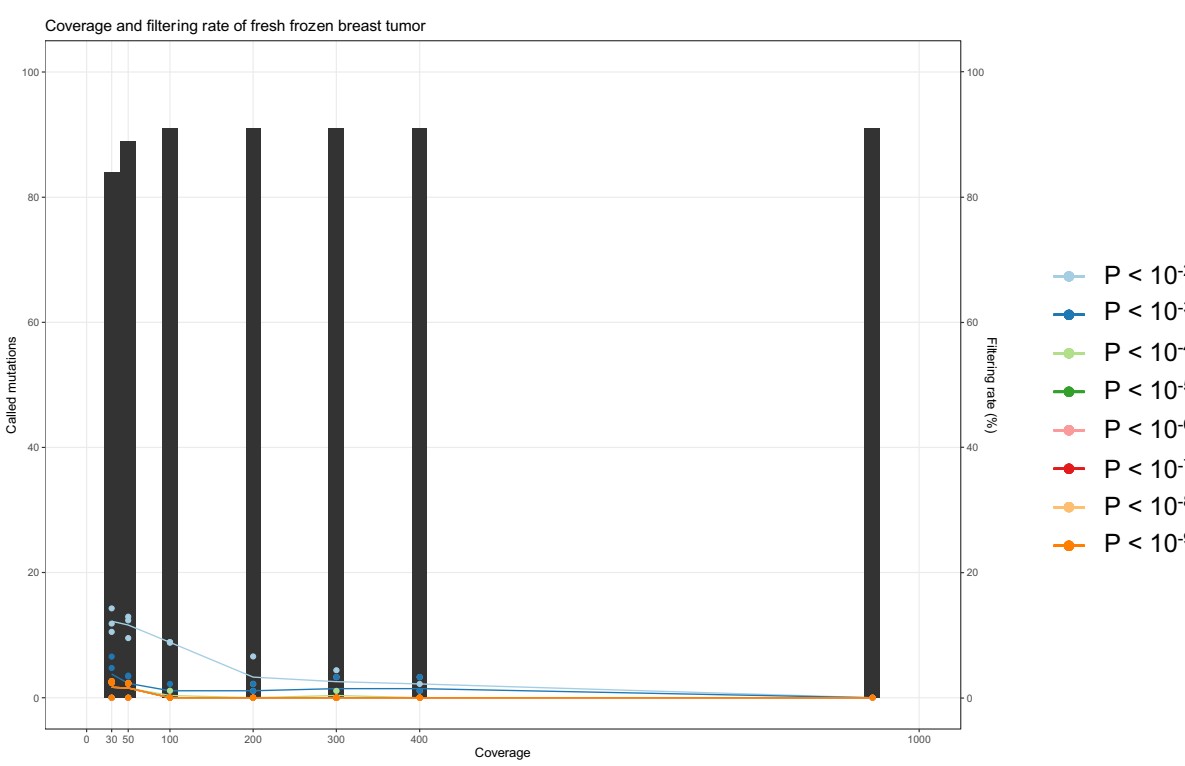

**Fig. 6 Appropriate mean coverage and _P_ value for MicroSEC.** The relationship between mean coverage, threshold of _P_ value, and artifact detection rate in FFPE normal breast tissue (**a**) and frozen breast tumor (**b**) are shown. From each library, three random sampling runs were performed so that the mean coverage of each library was 400, 300, 200, 100, 50, and 30. The average artifact detection rates (lines) and called mutation counts were calculated. The _P_ value thresholds for Filters 1–3 were varied from $10^{-2}$ to $10^{-9}$ to determine the appropriate threshold. The artifact detection rate decreased, and the number of false positives increased when the average depth of coverage was <100. In addition, false positives were seen in frozen tumor samples when the _P_ value threshold was $>10^{-5}$. FFPE formalin fixed and paraffin embedded.

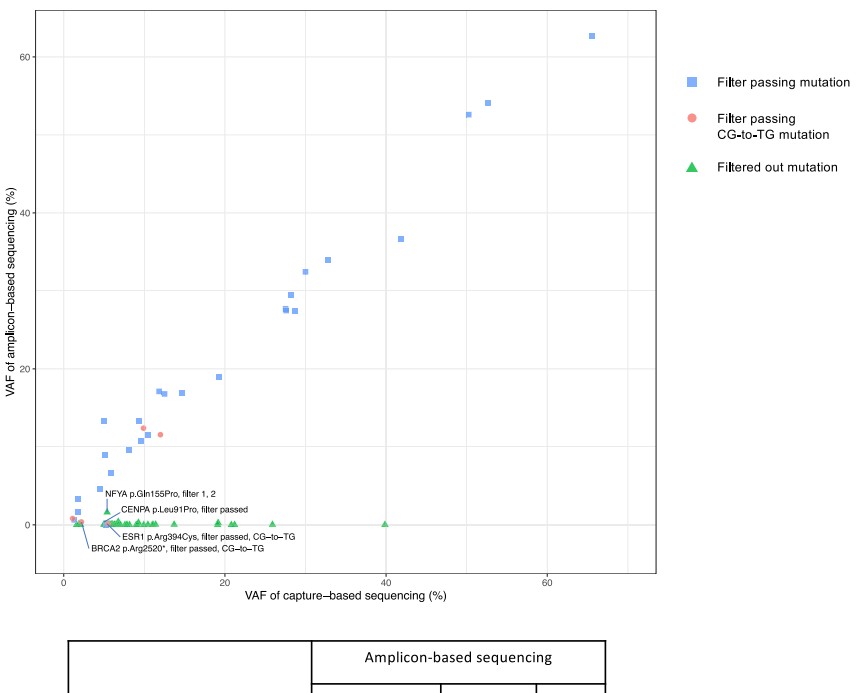

| | | Amplicon-based sequencing | | |
|---|---|---|---|---|
| | | True mutations | Artifacts | Total |
| MicroSEC Prediction | True mutations | 28 | 3 | 31 |
| | Artifacts | 1 | 65 | 66 |
| | Total | 29 | 68 | 97 |

**Fig. 7 Mutation validation by amplicon-based sequencing.** The MicroSEC analysis results were validated with amplicon-based sequencing that enriches target genomic regions by PCR. Ninety-seven mutations were randomly selected from 31 FFPE normal breast tissue samples, 12 FFPE breast tumor samples, 2 fresh frozen normal breast tissue samples, and 6 fresh frozen breast tumor samples. Each shape shows the variant allele frequencies (VAFs) in amplicon-based sequencing and capture-based sequencing for a specific mutation detected in a sample. The mutations that passed through the MicroSEC filter were detected with a similar level of VAF by both capture-based sequencing and amplicon-based sequencing (blue), with the exception of a *CENPA* mutation. Of the five potential CG-to-TG artifacts (red), two mutations in *ESR1* or *BRCA2* were not amplified by amplicon-based sequencing. Filtered out mutations were not detected by amplicon-based sequencing (green), with the exception of a *NFYA* mutation.

samples as artifacts, whereas MicroSEC detected 1199 SNVs as artifacts. A few pipelines are constructed to remove the chimeric reads generated by PCR, only during the sequencing of about 1.5 kilobase pairs of the 16S ribosomal RNA[14, 18, 19].

In contrast, MicroSEC utilizes such overlooked information. MicroSEC's focus on the distribution of mutations in each read enables it to remove only FFPE artifacts without eliminating the true mutations. The effectiveness of MicroSEC is similar to the experimental reduction of FFPE artifacts by degrading ssDNA with S1 nuclease[6]. MicroSEC is useful in applications in target deep sequencing or whole-exome sequencing data with very high coverage. A caveat of MicroSEC lies in its basis on probability calculations; therefore, it requires a high number of reads supporting each mutation. Overall, our pipeline will increase the reliability of the studies that use FFPE samples, thus advancing cancer research substantially. Our algorithm can also help to convey the correct information to cancer patients who underwent clinical sequencing, resulting in the improvement of quality and precision of their treatments.

## Methods
**Specimens**. The study cohort comprised 26 patients with breast cancer who underwent tumor resection or prophylactic surgery because of hereditary breast and ovarian cancer syndrome at St. Luke's International Hospital between 2010 and 2020 and 14 patients with primary cancer at National Cancer Center Hospital. A pathologist (N.K.) specializing in breast cancer reviewed the histological features and location of the tumors. Fifty-six FF samples of normal breast tissue were obtained from 14 patients, and 211 FFPE samples were obtained from 26 patients. Twenty-three FF breast cancer samples were obtained from 9 patients, and

37 FFPE samples were obtained from 11 patients, including matching FF and FFPE samples from 8 patients. We also obtained 14 matched FF and FFPE primary cancer samples from 12 colorectal adenocarcinoma patients and 2 oral squamous cell carcinoma patients. The surgeries were performed between 2012 and 2019. Tissue samples were provided by the National Cancer Center Biobank, Japan. The FFPE specimens were fixed in 10% neutral-buffered formalin for 15–72 h and then embedded in paraffin blocks. All FFPE specimens had been stored at room temperature for at least 6 months before DNA extraction. Blood samples were obtained from all the patients as a source of matching normal DNA. The study protocol was approved by the Ethics Committees of St Luke's International Hospital, National Cancer Center Hospital, and the National Cancer Center Research Institute. Written informed consent was obtained from all the participants.

**Capture-based panel sequencing and whole-exome sequencing**. Breast samples were subjected to capture-based panel sequencing, and primary cancer samples were subjected to whole-exome sequencing. Genomic DNA was isolated from FF or blood samples using QIAamp DNA Mini Kits (Qiagen, Germany) or FFPE samples using GeneRead DNA FFPE Kits (Qiagen, Germany). Then uracil DNA glycosylase treatment was performed for FFPE samples to remove C-to-T artifacts. Lastly, 500 or 50 ng of genomic DNA was subjected to target fragment enrichment using SureSelectXT Custom kits (Agilent Technologies, USA) or Twist Custom Panels (Twist Bioscience, USA) for capture-based panel sequencing, respectively. Libraries for whole-exome sequencing were generated from genomic DNA (50 ng) using the Twist Library Preparation EF kit and Twist Universal adapter system (Twist Bioscience). Enrichment of exonic fragments was performed by using Twist Human Comprehensive Exome Panel kit and Twist Fast Hybridization and wash kit (Twist Bioscience).

Custom-made probes were designed to hybridize and capture the target genes listed in the Todai OncoPanel (TOP)[2]. Custom-made probes for the panel hybridized and captured the genomic DNA of the target exons of 478 cancer-related genes. Because MGI sequencing platforms utilized single-stranded circular DNA libraries, adapter conversion PCR amplification was performed before sequencing. Double-stranded linear DNA libraries for Illumina sequencer were

**a**

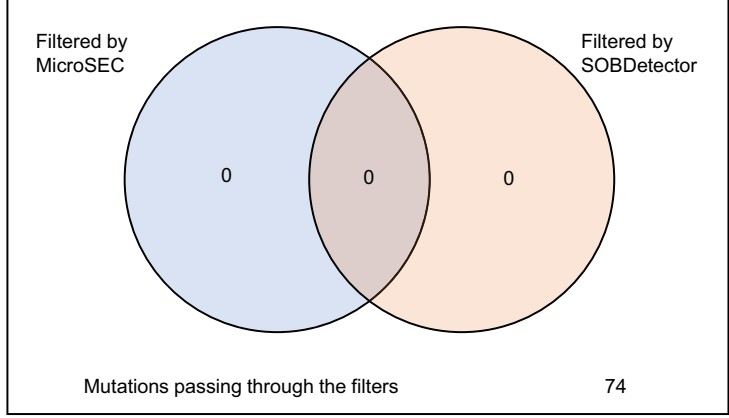

**b**

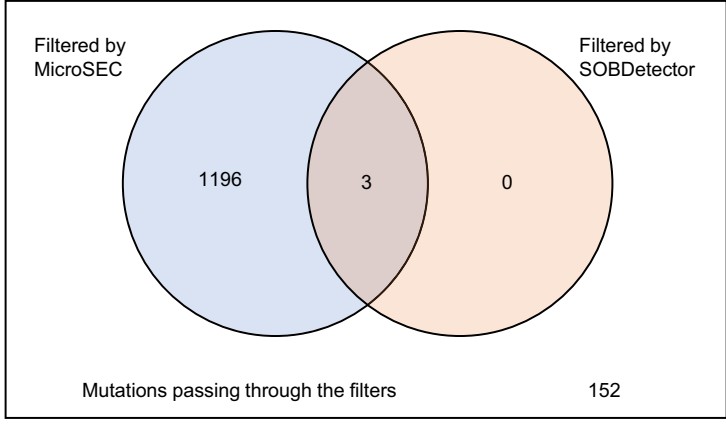

**Fig. 8 Filtering efficiency of SOBDetector and MicroSEC. a** The single-nucleotide variants detected in the frozen breast tumor samples were subjected to SOBDetector and MicroSEC. All 74 variants passed through both the filters. **b** The single-nucleotide variants detected in the FFPE normal breast tissue samples were subjected to SOBDetector and MicroSEC. Of the 1351variants, only 3 variants were filtered out by SOBDetector, whereas MicroSEC detected 1199 variants as possible artifacts. FFPE formalin fixed and paraffin embedded.

converted to single-stranded circular DNA libraries for MGI sequencer with the MGIEasy Universal Library Conversion Kit (MGI Technologies, China). A library's quantity and quality were assessed with a Qubit 2.0 Fluorometer (Thermo Fisher Scientific, USA) and the Agilent 2200 TapeStation System (Agilent Technologies, USA). The massive parallel sequencing of the isolated fragments with 124–151 base pair read lengths was performed using the HiSeq 2500 System, NovaSeq 6000 System (Illumina, USA), or DNBSEQ-G400 sequencer (MGI Technologies, China). Samples with an insufficient mean coverage of <400 were excluded ($n = 28$, 8.6%) in capture-based panel sequencing.

**Amplicon-based sequencing**. Ninety-seven mutations were randomly selected so that the ratio of true mutations to artifacts predicted by MicroSEC was approximately 1:2. The mutations were derived from 31 FFPE normal breast tissue samples, 12 FFPE breast tumor samples, 2 FF normal breast tissue samples, and 6 FF breast tumor samples. The genomic regions of 101–156 bases around a mutation were amplified by PCR using the NEB Q5 Hot Start HiFi PCR Master Mix (New England BioLabs, USA) and appropriate primer sets (Supplementary Data 1). PCR products were purified using Agencourt AMPure XP beads (Beckman Coulter, USA) and subjected to library preparation with the NEBNext Ultra II DNA Library Prep Kit for Illumina (New England BioLabs, USA). The libraries were sequenced to generate 150 base paired-end reads using the MiSeq system (Illumina, USA). If the VAF of a mutation by amplicon-based sequencing was <30% of the VAF of the

mutation by capture-based sequencing, we determined that the mutation was an artifact.

**Somatic mutation calls**. Paired-end reads were independently aligned to the human reference genome (University of California, Santa Cruz Genome Browser assembly ID: hg38) using Burrows-Wheeler Aligner (v0.7.17)[20] and Bowtie2 (v2.1.0)[21]. Potential PCR duplicates were removed with SAMtools (v1.9)[22]. Since a single algorithm might fail to detect important mutations, the union of somatic mutations identified using MuTect2 (GATK v4.1.3.0)[23], VarScan2 (v2.4.3)[16], and our in-house pipeline was used for the analysis. Somatic mutations in each sample were determined by comparison with sequence data from normal blood from the same individuals.

Mutations were discarded if the read depth was <100 base pairs or the number of mutation-supporting reads was <10 or VAF was <5%. Mutations were also eliminated if the reads were supported by only one strand of the chromosome, observed in normal human genomes in the 1000 Genomes Project dataset[24] with a frequency >1% or in our in-house database, located outside the 3.4 megabase TOP target region, or located at simple repeat sequences downloaded from the Tandem Repeats Database (https://tandem.bu.edu/cgi-bin/trdb/trdb.exe)[25, 26]. In addition, the mutations were removed if a homopolymer consisting of >14 bases was present within 20 bases around the mutation; or >10% of the bases within the mutation supporting reads had a Phred quality score of <Q18. The mapped reads and called

mutations were visualized and manually checked with the Integrative Genomics Viewer (v2.4.10)[27].

**MicroSEC pipeline**. MicroSEC (v1.2.8) is a filtering pipeline written in R language designed to discover MICR-derived sequencing errors in FFPE samples. It detects local palindromic sequences, the uneven distribution of mutated bases within each read, and the pseudo-mutations introduced by distant homologous regions.

The workflow of MicroSEC in this study was composed of several steps. First, the positions of the mutated bases were detected with the matchPattern function from the Biostrings package (v2.54.0). Second, the adapter sequence at the 3′ end of each read was trimmed with the trimLRPatterns function from the Biostrings package with the maximum mismatch rate of 10%, so as to analyze libraries that did not perform adapter trimming in the quality check step. Then, the distance from the altered base to the read's effective ends was calculated based on the Compact Idiosyncratic Gapped Alignment Report (CIGAR) strings of the alignment results (see "Supporting length analysis"). Next, the hairpin structure-induced chimeric reads were detected (see "Hairpin structure detection"). Afterward, the entire genome was searched for regions that were homologous to the sequence around the mutation using the countPDict function from the Biostrings package (see "Homologous region detection"). Lastly, the above results were combined to construct the MicroSEC filtering pipeline. The results of MicroSEC's filtering of the somatic mutations in breast tissue samples and clinical sequencing samples were provided as Supplementary Data 1.

**Supporting length analysis**. It is essential to identify the uneven distribution of the mutations in the mutation-supporting reads for discriminating between the MICR-derived errors and true mutations. Here we have introduced the concept of support lengths, which are the base lengths from the mutated bases (excluding the mutated bases) to the furthest mapped bases in each read, based on the CIGAR strings in BAM files (Fig. 3). When a mutation is a substitution of a number ($N$) of nucleotides, and the read length is another number ($L$) of nucleotides, the 3′- or 5′-supporting lengths are distributed between zero and $L − N$. When a mutation is an $N$-nucleotide deletion or insertion, the possibility of the mutation being the part of a repeated sequence should be considered, because the mutation-supporting reads must contain all the repetitive sequence to determine the presence or absence of small insertions/deletions (indels).

Furthermore, we need to consider the penalties by Burrows–Wheeler Aligner for mapping. Since the penalty due to an $N$-base mutation is $N + 6$, the soft-clipping penalty is 5, and the point for mapped $M$ bases is $M$, a sequence supporting the mutation would be soft-clipped if there are no more than $N + 1$ matching sequences outside the gap (Supplementary Fig. 5a). Given the 5′ and 3′ repetitive sequences around the mutation of lengths $R_{5′}$ and $R_{3′}$, the supporting lengths can be distributed between $\max(R_{3′ \text{ or } 5′}, N + 1)$ and $L − \max(R_{5′ \text{ or } 3′}, N + 1)$ for deletions or between $\max(R_{3′ \text{ or } 5′}, N + 1)$ and $L − N − \max(R_{5′ \text{ or } 3′}, N + 1)$ for insertions, with $\max(X, Y)$ denoting the larger value of $X$ and $Y$ (Supplementary Fig. 5b).

We also have introduced the concept of the shorter support length, which is the shorter of the 3′- and 5′-supporting lengths in each read. The theoretical distributions are between zero and $[(L − N)/2]$, between $\min(\max(R_{3′}, N + 1), \max(R_{5′}, N + 1))$ and $[L/2]$, and between $\min(\max(R_{3′}, N + 1), \max(R_{5′}, N + 1))$ and $[(L − N)/2]$, for substitutions, deletions, and insertions, respectively. $[X]$ represents the greatest integer ≤$X$, with $\min(X, Y)$ denoting the smaller value of $X$ and $Y$.

Suppose the actual supporting length distribution is between $A$ and $B$, the theoretical distribution is between $C$ and $D$, and the coverage depth of the mutation-supporting reads is $E$. In that case, the probability $p$ is calculated based on the multinomial distribution by the following equations:

$$p = \left\{ \sum_{m=A}^{B} f(m) \Big/ \sum_{n=C}^{D} f(n) \right\}^{E} \tag{1}$$

$$f(x) = [\text{The number of reads of which the supporting length is } x.] \tag{2}$$

We regard mutations with $p < 10^{-6}$ as errors in principle. During deep sequencing, however, $p$ becomes too small with large $E$, leading to the overfiltering of the mutations. Thus, the filter is not applied to the mutations if $B − A + 1$ is >75% of $D − C + 1$ or $\sum_{m=A}^{m=B} f(m)$ is >75% of $\sum_{n=C}^{n=D} f(n)$.

When multiple mutations are close to each other, the minimum required supporting length will increase because of the cumulative mismatch penalties of the mutations. Overfiltering can be avoided by fixing the minimum supporting length to zero if there are more than three mismatched bases in the ten bases surrounding the mutation.

**Hairpin structure detection**. A sequence of at least 15 consecutive bases, with the mutated bases at the center, was extracted from each read. If the sequence was present on the opposite strand and within 200 bases of the neighboring sequence, the mutation was suspected to be derived from a hairpin structure. If >50% of the reads had such sequences, the mutation would be considered an error.

**Homologous region detection**. We extracted a sequence comprising the mutated bases and the surrounding repetitive sequences if available from each read. Four upstream or downstream bases were added to that sequence, and a sequence further from the sequence with the mutation was added on the opposite side to create 2 sequences with a total of 40 bases. We searched the entire genome for regions matching exactly to one of the two sequences. If >15% of the reads had such homologous sequences, the mutation would be considered an artifact.

**Random sampling**. From the library obtained by capture-based sequencing of each sample, random sampling was performed 3 times each to obtain average coverages of 400, 300, 200, 100, 50, and 30 using SAMtools. The artifact detection rate was calculated as the average of the results of the three random sampling.

**SOBDetector**. SOBDetector (v1.0.2) was applied to 74 SNVs detected in frozen breast tumor samples and 1351 SNVs were detected in FFPE normal breast tissue samples with default settings.

**Statistics and reproducibility**. Statistical analyses were performed with R version 4.0.5. The local regression curve in Fig. 4 and kernel density plots in Fig. 5b were drawn by ggplot2 package version 3.3.3. Sensitivity and specificity of MicroSEC was estimated with epiR package version 2.0.19.

**Reporting summary**. Further information on research design is available in the Nature Research Reporting Summary linked to this article.

## Data availability

Source data underlying all the figures and tables in the manuscript are available in Supplementary Data 1 and Supplementary Data 2. We have deposited the raw sequencing data of the breast tissue samples and primary cancer samples examined in this study under the accession number JGAS000368 and JGAS000377, respectively, in the Japanese Genotype-Phenotype Archive (http://trace.ddbj.nig.ac.jp/jga) hosted by the DNA Data Bank of Japan. The target sequencing data of 54 cancer specimens are available for download at the Japanese Genotype-Phenotype Archive under the accession number JGAS000164. All other data are available from the corresponding author on reasonable request.

## Code availability

All the source code written in R for MicroSEC is publicly available at Zenodo[28] and https://github.com/MANO-B/MicroSEC.

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

## Acknowledgements

The authors would like to thank A. Shiino for technical assistance. The study was supported in part by the World-leading Innovative Graduate Study Program for Life Science and Technology, The University of Tokyo, as part of the WISE Program (Doctoral Program for World-leading Innovative & Smart Education), MEXT, Japan; JSPS KAKENHI grants (#19J13207); Grant of Japan Orthopaedics and Traumatology Research Foundation (No. 418); and AMED under Grant Number JP19ck0106252. National Cancer Center Biobank is supported by the National Cancer Center Research and Development Fund, Japan.

## Author contributions

M.I. conceived the project and developed the algorithm. N.K. and H.Y. collected breast tissue samples. S.I., T.M., S.S., Y.I., and Y.Y. collected primary cancer tissue samples for whole-exome sequencing. M.I., S.K., T.H., and T.U. designed and analyzed the experiments. H.K. and S.T. supervised the projects. M.I., S.K., T.H., T.U., and H.M. wrote the manuscript with input from all authors. All authors have read and approved the manuscript.

## Competing interests

The authors declare no competing interests.
