## [Peer Review File · Communications Biology]

Reviewers' comments:

Reviewer #1 (Remarks to the Author):

Ikegami and colleagues provide a valuable tool, MicroSEC, for filtering FFPE sequencing artifacts. Archival FFPE tissue is of great interest for genomic analysis, but it is difficult to obtain reliable sequencing data from these samples. The authors provide a tool that may greatly increase the ability to analyze these tissues, making the study of significant value.

Major issues:

a) Performance of MicroSEC. Memory requirement is huge making the tool almost unusable in analyzing real tumor samples. We tried on mutations generated FFPE breast cancer exome sequencing and was not able to complete to only 10 mutations by requesting 300GB of memory on our high-performance computing cluster. The job failed as it exceeded allocated 300GB memory usage. Breast cancer exome data can have >100 somatic mutations so the software is not capable of running the real clinical samples with the exception of those generated by a very small gene panel. This shows that the tool is incredibly inefficient and will require major enhancement to improve the memory usage. Otherwise, MicroSEC is not easily scalable for analyzing exome, which will greatly limit its applicability.

b) The authors note that MicroSEC is best used in applications with high coverage, such as targeted deep sequencing or exome sequencing. Why is a large read depth required to run MicroSEC? None of the filtering criteria listed in Fig. 1d are obviously related to sequencing depth. For example, the identification of palindromes near mutated sites can be readily done regardless of sequencing depth. The authors need to present data to demonstrate the effect of sequence coverage on filtering the artifacts of FFPE sample. It would also be good to expand the discussion so that users interested in analysis of FFPE WGS data can better understand the applicability of MicroSEC on FFPE WGS which tends to be lower coverage.

Minor issues for improving the clarity of the manuscript:

1. It would be helpful if Fig. 1, panels a-c could each be explained in detail, one-by-one, in the manuscript text. Each of these panels contains extensive information but the panels are referenced all at once and only briefly. For example, what do the green and blue arrows represent in Fig. 1b, c? Are these PCR amplification steps? During which step of sequencing? What do the colored (blue, gold) reads represent in Fig. 1a? Also, the Fig. 1a legend mentions soft-clipping near the FGFR4 mutation, and it would be helpful to actually show the soft-clipped portions of the reads.
2. Likewise, in Fig. 1d several features are not defined. What do the vertical green bars represent? Under "Filter 4", what do the green and dark-red reads represent? Please add text to describe each of these (e.g. "insert", "distant homologous region", etc.) to aid interpretation. Additionally, if the authors can clearly differentiate between sequencing reads and inserts/fragments that would be helpful.
3. Does Fig. 1e represent a single sample or the aggregate of multiple samples?
4. The authors note that both MuTect and VarScan were used for variant calling. Was the intersection or union of variants used?
5. In the manuscript, targeted capture sequencing is employed. How many genes or sites were analyzed per sample (using the OncoPanel described in Methods)? What was the approximate length in base pairs of each captured region? A brief description of the OncoPanel analyzed would be helpful.
6. In Supplementary Fig. 2a, variants from normal breast tissues were analyzed. Was this paired breast vs. blood analysis to detect somatic variants in the normal breast tissue? Please also indicate the number of mutations represented by each panel in Supplementary Fig. 2.
7. It would be helpful if the authors could show an application of MicroSEC on FFPE exome or FFPE WGS data, which would greatly expand the utility of the tool.
8. The authors look for palindromes "within 150 bases of the neighboring sequence." What was the insert length used by the authors for capture sequencing? Relatedly, does MicroSEC look for palindromes/homology within sequencing inserts, individual reads, or both?
9. The authors state " $p < 10^{-6}$ are considered artifacts". How was this cutoff determined?
10. The authors state "VarScan2 mutation caller employs it as a filtering." Could the authors clarify

what "it" refers to?

11. Does MicroSEC act on BAM files? Is a certain type of alignment required? A brief (1-2 sentence) technical explanation would be helpful in the manuscript text.

12. In Supplementary Fig. 4, the authors show 56 FFPE-specific variants which were not filtered out by MicroSEC. What would be the plausible explanation for these artifacts? Could the authors suggest approaches that might filter these variants?

The main result of this paper is that using the filtering algorithm pipeline (MicroSEC) that they developed could potentially filter formalin-fixed and paraffin-embedded (FFPE) mutational artefacts from post hoc DNA sequencing. The lack of availability of fresh or fresh frozen (FF) samples leads to the use of FFPE samples for sequencing purposes, resulting in a high number of variants calling errors. FFPE variant calling errors are the result of material degradation and are characterized by a high number of single-stranded DNA (ssDNA). The paper also claims that microhomology induced chimeric read (MICR) due to ssDNA is the major cause of error-prone sequencing aside from the well-known CG-to-TG mutational artefacts patterns in FFPE.

In their paper, the authors have tested their theory, performing their filtering pipeline approach in breast cancer and normal breast tissue with some independent and some matching FF and FFPE samples. A weak but understandable point of the filtering usage is the requirement of at least 400-reads mean coverage. Since the authors used their method in target sequencing data, the requirement could be met, but this would not be as easy to accomplish in common whole-genome sequencing. However, the approach could potentially be useful in gene panel sequencing as they demonstrated. As could be expected from a filtering tool, the paper method points out two important facts. First is that mutations which passed through the filter tended to have a higher variant allele frequency (VAF), and second, that the MicroSEC did not remove any mutation detected in matched FF samples. Both are expected in a well-working tool for filtering artefacts and selecting those real mutations.

I found the method reliable, making use of many thoughtful and deliberated steps to account for many different possible scenarios for filtering artefacts; here are some considerations to add to my perspective:

1.- The most important considerations I see is that the paper claims there are some methods already available for artefact filtering. However, they are “not highly accurate” (line 148). The question is, what is the basis of this claim? No reference is given here, and most importantly, no comparison is made between the performance of their method and those of the “not highly accurate” algorithms on uniform data, mutations, and filtering artefacts. This should be taken into account.

2.- Predictions were made based on 11 mutations that passed through the MicroSEC filtering, and 14 were not detected. The authors should elaborate more fully as to whether 11 detected and 14 non-detected mutations are reliable enough results to determine the success of their method. Are they enough?

3.- Regarding the MicroSEC pipeline: it's not clear, and I don't find any reasonable excuse for performing the adapter sequence trimming at the second step instead of doing it during the first step as part of a normal QC process. And once the reads pass the QC steps, the pipeline could start with the position of the mutated bases that have been detected. If there is a reason for this ordering, the authors should explain it, as it is not clear.

*4. I also had some concerns about some of the figures. Starting from **Supplementary Figure 2**. As I understand, Top 2a and Top 2b represent the total number of mutations detected, with their corresponding VAF. How is it possible that the y-axis figure a goes up to 900*

mutations, with the highest bar not reaching 1000, while in the “Filtered out mutations (normal, FFPE)” (which suggests that it should include fewer mutations) is also nearly 1000 mutations? Also, in “Mutations passing the filter (normal, FFPE),” the max bar is over 150 mutations. The sum of “Filtered out mutations (normal, FFPE)” and “Mutations passing the filter (normal, FFPE)” should be equal to “Total mutations (normal, FFPE),” and clearly the sum is over that. In the figure b, the numbers seem reasonable, but not in a. **Supplementary Figure 4.** Same question as in the point 2. Are those numbers enough to determine the successful of the method? **Supplementary Figure 5.** Regarding the appropriate thresholds confirmed in this figure: I can see in Figure 5a that the threshold could be appropriated for filtering purposes, but certainly in 5c and 5d, that threshold in 50% and 15% respectively could be improved. And more explanation about why are appropriated could be helpful.

5. Lastly, there is a lack of clarity starting from line 251 in the Supporting length analysis that makes it difficult to understand the method. I can't give any strong opinion if the mathematical method is robust as I don't get the meaning of it. I became completely lost beginning with “no more than $N+1$ matching sequences outside the gap”, which I think should be clarified. The same thing happens on line 150, causing things to be unclear.

Overall, the idea and goal are clear, and steps are thoughtfully selected. The method could potentially influence certain future sequencing perspective as it paves a way to reanalyse older data or new FFPE samples which were never used due to artefacts problems. However, the paper needs further comparison with available tools to demonstrate exactly what their improvement is.

We would like to thank the two reviewers for the thoughtful comments. We have revised and modified the manuscript in accordance with the reviewers' comments and suggestions, and carefully proofed the manuscript to remove all typographical and grammatical errors. We believe that this revised version is greatly improved due to the clear and helpful comment from both reviewers.

Find below a point-by-point response to the concerns raised by each reviewer.

Reviewer #1 (Remarks to the Author):

Ikegami and colleagues provide a valuable tool, MicroSEC, for filtering FFPE sequencing artifacts. Archival FFPE tissue is of great interest for genomic analysis, but it is difficult to obtain reliable sequencing data from these samples. The authors provide a tool that may greatly increase the ability to analyze these tissues, making the study of significant value.

Major issues:

- a) Performance of MicroSEC. Memory requirement is huge making the tool almost unusable in analyzing real tumor samples. We tried on mutations generated FFPE breast cancer exome sequencing and was not able to complete to only 10 mutations by requesting 300GB of memory on our high-performance computing cluster. The job failed as it exceeded allocated 300GB memory usage. Breast cancer exome data can have >100 somatic mutations so the software is not capable of running the real clinical samples with the exception of those generated by a very small gene panel. This shows that the tool is incredibly inefficient and will require major enhancement to improve the memory usage. Otherwise, MicroSEC is not easily scalable for analyzing exome, which will greatly limit its applicability.

The reviewer pointed out a crucial problem of MicroSEC. We designed MicroSEC for analysis of targeted sequencing, and the previous version of MicroSEC (version 1.1.18) required lots of memory for whole exome or whole genome sequencing. We added a pre-processing step to MicroSEC to remove reads irrelevant to the mutation from the BAM file, which reduced the memory requirement and processing time (requesting 1.5 gigabytes of memory). Currently, MicroSEC version 1.2.7 can process 4,707 of mutations of target sequencing data used in this article in six hours on a MacBook Air with 16 gigabytes of memory. Along with the minor comment #11, we have added this performance in the Result section. Due to the upgrade of MicroSEC, the results of Fig. S3 are now only available for mutation depth 10 and above. The figure was modified by adding a local regression curve.

Fig. S3 was modified.

Supplementary Figure 3. The relationship between the mutation depth and filtering efficiency in the 190 FFPE normal breast tissue samples.

The histograms of mutation depth for total mutations (gray) and mutation passing the MicroSEC filter (blue) were shown. Most of the called mutations are considered to be artifacts as normal breast tissues have little or no mutations. The mutation filtering rate (black line) is high at low depth and more than 90% of mutations are filtered by MicroSEC pipeline when the mutation coverage is 10–22. There is a tendency for the percentage of reads that passed the filter to increase as the number of reads with mutations increases. Local regression curve of the filtering rate is shown as a red line. FFPE, formalin-fixed and paraffin-embedded.

The following sentences were added.

Lines 109–115: Based on our theory that artifacts are derived from ssDNA-annealing, we have developed a MICR-originating Sequence Error Cleaning pipeline (MicroSEC), a post hoc filtering pipeline to predict if a given mutation is an MICR-derived error. This pipeline allows the processing of thousands of mutations of target sequencing data within hours on a standard PC with 16 gigabytes of memory. MicroSEC requires a list of mutations and corresponding BAM files, rather than FASTQ files as it uses the positional bias of reads mapped against mutations.

Lines 148–153: The relationship between the filtering rate and mutation coverage was also examined using FFPE samples from normal breast tissues, which were expected to have few or no mutations. More than 90% of mutations were filtered by MicroSEC pipeline when the mutation coverage was 10–22. A tendency to increase could be observed for the mutation filtering rate as the number of reads with mutations increased (Supplementary Fig. 3).

- b) The authors note that MicroSEC is best used in applications with high coverage, such as targeted deep sequencing or exome sequencing. Why is a large read depth required to run MicroSEC? None of the filtering criteria listed in Fig. 1d are obviously related to sequencing depth. For example, the identification of palindromes near mutated sites can be readily done regardless of sequencing depth. The authors need to present data to demonstrate the effect of sequence coverage on filtering the artifacts of FFPE sample. It would also be good to expand the discussion so that users interested in analysis of FFPE WGS data can better understand the applicability of MicroSEC on FFPE WGS which tends to be lower coverage.

Thank you for the comment.

- Figure 1d illustrates only the concept of filtering process of ssDNA-derived artifacts, and the figure does not show the exact filtering criteria. Therefore, Figure 2 was added to explain the details of filtering process by MicroSEC.

- The detection of palindromes in the vicinity of a mutation requires that at least 50% of the mutation supporting reads actually contain the palindromic sequence, so the results are not trustworthy unless there are a certain number of mutation supporting reads. For a mutation with 5% VAF, the total number of reads must be more than 200 for there to be 10 mutation supporting reads.

- To evaluate the relationship between mean coverage and artifact detection rate, we performed random sampling from sequences of FFPE samples of normal breast tissue and frozen samples of breast cancer. From each library, three random sampling runs were performed so that the mean coverage of each library was 400, 300, 200, 100, 50, and 30. We calculated the average artifact detection rate after excluding mutations for which there were no reads observed. We also varied the p-value threshold for Filter 1 and Filter 3 from 10^{-2} to 10^{-9} to determine the appropriate threshold. The results are shown in Fig. S7. The results showed that the detection rate of artifacts decreased, and the number of false positives increased when the average depth of coverage was less than 100. In addition, false positives were seen in frozen tumor samples when the p-value threshold was above 10^{-5} , indicating that the threshold should be below or equal to 10^{-5} . The threshold can be 10^{-5} , but to avoid over-filtering we adopted 10^{-6} .

- We could not obtain WGS data of FFPE samples, so we added a comparison of WES analysis of matched frozen and FFPE samples from 14 primary cancers. The results are shown in Fig. S4b and Table S3.

Fig. 2, Fig. S4b, and Fig. S7 were added.

Figure 2.

Figure 2. The details of the MicroSEC filtering criteria.

a. Definition of supporting lengths. Supporting lengths are defined as the distances from the mutated base to the 5' or 3' ends of an individual read (excluding soft-clipped bases). The shorter supporting length is defined as the shorter one.

b. Filtering based on biased distribution of supporting lengths. The supporting lengths are calculated only for the reads with a mutation. A and B indicate the minimum and maximum lengths, respectively. The supporting lengths that the mapping software can theoretically generate for reads with the mutation are determined. C and D indicate the minimum and maximum values, respectively. E indicates the total count of reads with mutations. Based on the distribution of supporting lengths in reads without mutations, the probability (p_0) that the supporting lengths are between A and B is calculated (equation (1)).

c. Filtering based on suspected hairpin formation. Based on the putative mechanism of artifact generation during the end-repair step, the sequence around the mutation is derived from the opposing strand.

Sequences of 15 bases containing the mutation are extracted from each read and mapped to the opposite strand within 200 bases of the neighboring sequence. If a 15-base sequence is mapped without mismatch, the read is considered to be hairpin-derived. The mutation is considered as an artifact if more than half of the reads are hairpin-derived. ssDNA, single-stranded DNA.

d. Filtering for distant homologous region-derived artifacts. A G>T artifact in chromosome 19 is shown. Sequences of 40 bases containing the mutation are extracted from each read. The mutation is considered an artifact if more than 15% of the 40-base sequences match completely to other regions in the genome.

a

b

Whole exome sequencing on matched samples of primary cancer

Supplementary Figure 4. The mutations detected in matching FF and FFPE samples.

a. The somatic mutations detected in the target sequencing of matching sets of frozen (blue) and FFPE (peach) breast cancer samples from eight patients. Eleven mutations found in only FFPE samples were filtered by MicroSEC (green). All mutations found in fresh frozen samples passed through the MicroSEC filter. b. The somatic mutations detected in the whole exome sequencing of matching sets of frozen (blue) and FFPE (peach) primary cancer samples from 14 patients. The 123 mutations (21.0%) found in only FFPE samples were filtered by MicroSEC (green). Eight mutations (0.5%) found in fresh frozen samples were filtered by the MicroSEC filter. FFPE, formalin-fixed and paraffin-embedded.

Supplementary Figure 7. Appropriate mean coverage and P-value for MicroSEC.

The relationship between mean coverage, threshold of P-value, and artifact detection rate in FFPE normal breast tissue (a) and frozen breast tumor (b) are shown. From each library, three random sampling runs were performed so that the mean coverage of each library was 400, 300, 200, 100, 50, and 30. The average

artifact detection rates (lines) and called mutation counts were calculated. The P-value thresholds for Filters 1 and 3 were varied from 10^{-2} to 10^{-9} to determine the appropriate threshold. The artifact detection rate decreased, and the number of false positives increased when the average depth of coverage was less than 100. In addition, false positives were seen in frozen tumor samples when the P-value threshold was above 10^{-5} . FFPE, formalin-fixed and paraffin-embedded.

The following sentences were added.

Lines 122–123: The shorter of the 3' and 5'-supporting lengths is defined as the shorter supporting length.

Lines 125–128: During the step of detecting hairpin-induced errors, 15-base sequences containing the mutation were extracted from each read (Fig. 2c). We considered the mutation an artifact if more than half of the sequences existed on the opposite strand within 200 bases of the neighboring sequence.

Lines 188–198: To evaluate the relationship between the mean coverage and artifact detection rate, we performed random sampling from the sequences of FFPE normal breast tissue samples and frozen breast cancer samples. From each library, three random sampling runs were performed so that the mean coverage of each library was 400, 300, 200, 100, 50, and 30. We calculated the average artifact detection rate after excluding mutations for which there were no reads observed. We also varied the P-value threshold for Filters 1 and 3 from 10^{-2} to 10^{-9} to determine the appropriate threshold (Supplementary Fig. 7). The artifact detection rate decreased, and the number of false positives increased when the average depth of coverage was less than 100. In addition, false positives were seen in frozen tumor samples when the P-value threshold was above 10^{-5} , indicating that the threshold should be below or equal to 10^{-5} . Although the threshold could be 10^{-5} , we adopted 10^{-6} to avoid overfiltering.

Lines 228–238: We investigated the applicability of MicroSEC not only for target deep sequencing but also for whole exome sequencing with a relatively lower coverage. Whole exome sequencing data of 14 matched FF and FFPE samples of primary cancer (12 colorectal adenocarcinoma and two oral squamous cell carcinoma) was performed with a mean coverage of 199 and 255, respectively (Supplementary Table 3). Our mutation caller identified 107.0 and 118.2 mutations per sample in the FF and FFPE samples, respectively. Only 0.6 mutations per sample (0.5%) detected in the FF samples were filtered by the filter, while 10.3 mutations per sample (8.7%) were filtered out in the FFPE samples (Supplementary Fig. 4b). This result suggests that the ssDNA derived artifacts are also present in the sequencing data of frozen samples. With sufficient coverage, MicroSEC was considered to be applicable for whole exome sequencing.

Lines 269–272: The study cohort comprised 26 patients with breast cancer who underwent tumor resection or prophylactic surgery because of hereditary breast and ovarian cancer syndrome at St. Luke's International Hospital between 2010 and 2020, and 14 patients with primary cancer at National Cancer Center Hospital.

Lines 277–280: We also obtained 14 matched FF and FFPE primary cancer samples from 12 colorectal adenocarcinoma patients and two oral squamous cell carcinoma patients. The surgeries were performed between 2012 and 2019. Tissue samples were provided by the National Cancer Center Biobank, Japan.

Lines 409–413: **Random Sampling** From the library obtained by capture-based sequencing of each sample, random sampling was performed three times each to obtain average coverage of 400, 300, 200, 100, 50, and 30 using SAMtools. The artifact detection rate was calculated as the average of the results of the three random sampling.

Minor issues for improving the clarity of the manuscript:

1. It would be helpful if Fig. 1, panels a-c could each be explained in detail, one-by-one, in the manuscript text. Each of these panels contains extensive information but the panels are referenced all at once and only briefly. For example, what do the green and blue arrows represent in Fig. 1b, c? Are these PCR amplification steps? During which step of sequencing? What do the colored (blue, gold) reads represent in Fig. 1a? Also, the Fig. 1a legend mentions soft-clipping near the FGFR4 mutation, and it would be helpful to actually show the soft-clipped portions of the reads.

Thank you for the comment. Fig. 1a-c were not explained well enough, so we added detailed descriptions to the main text and the legend.

- The blue arrow in Fig 1b, c represents the upstream sequence of the specific read with artifact in *FGFR4* gene, and the green arrow represents the downstream sequence of the same read.
- Artifacts are generated at the end-repair step of library preparation.
- The blue colored read in Fig. 1a has an inferred insert size smaller than expected.
- The mate-reads of green or gold colored reads in Fig. 1a are mapped to different chromosomes.
- The soft-clipped portion of the reads are indicated by a red line in Fig. 1a.

Fig. 1 was corrected.

Figure 1.

Figure 1. An example of microhomology-induced chimeric read (MICR)-originated sequencing error.

- a.** The genomic sequence visualized by Integrative Genomics Viewer exhibits a T-to-C artifact in the *FGFR4* gene. In all mutation-supporting reads, only six bases downstream of the mutation were mapped, and the rest is soft-clipped (red line). The blue colored read has an inferred insert size smaller than expected. The mate-reads of green or gold colored reads were mapped to different chromosomes.
- b.** A representative read supporting the T-to-C artifact in Fig. 1a. The upstream sequence of the read (blue arrow) was mapped to the forward strand of the genome, and the downstream sequence of the same read (green arrow) was mapped to the reverse strand. Two palindromic sequences exist in close proximity to each other, and the mismatched base between the two sequences (red box) represent the source of the T-to-C artifact. Most of the downstream bases were soft-clipped.
- c.** Two palindromic sequences in a single-stranded DNA (ssDNA) formed a hairpin structure at the end-repair step of library preparation. After nicking and partial denaturation, the double-stranded DNA was regenerated during the end-repair step of library preparation. The mismatched base between two palindromic sequences was defined as a mutation.
- d.** The MicroSEC algorithm is based on three criteria. Filter 1, 3: the distance from the mutation position to the most distant mapped base is distributed over a probabilistically improbable limited range for any reads. Filter 2: MICR-originated sequencing errors are generated when two palindromic sequences are in the same DNA fragment. Filter 4: The mis-annealing of ssDNA derived from other distant homologous regions of the genome also creates chimeric reads and artifacts. Dark-red, green, or light-blue horizontal bars represent sequences of other distant regions of the genome. Chimeric reads with mutated bases were formed.

The following sentences were added.

Lines 75–89: First, we found mapping anomalies characteristic of artifacts in FFPE samples. DNA extracted from samples was fragmented at random positions to the appropriate size before sequencing. Mutated bases are expected to be distributed evenly in the reads. However, a marked bias in the position of the mutation was observed in the case of a T-to-C artifact in the *FGFR4* gene (Fig. 1a). In the case of all reads with the artifact, only six bases downstream of the mutation were mapped, and the rest were soft-clipped. This phenomenon was not observed in the non-mutated reads. The mapping of a read with an artifact in *FGFR4* was examined in detail (Fig. 1b). The upstream sequence of the read was mapped to the forward strand of the genome, and the downstream sequence was mapped to the reverse strand of the same genomic region. Two palindromic sequences exist in close proximity to each other in this region. From this, we estimated the phenomena shown in Fig. 1c in the end-preparation step of library preparation. A ssDNA containing two palindromic sequences potentially formed a hairpin structure. After nicking and partial denaturation, the double-stranded DNA could be regenerated by DNA polymerase. Then, the mismatched base between two palindromic sequences was detected as a mutation.

2. Likewise, in Fig. 1d several features are not defined. What do the vertical green bars represent? Under “Filter 4”, what do the green and dark-red reads represent? Please add text to describe each of these (e.g. “insert”, “distant homologous region”, etc.) to aid interpretation. Additionally, if the authors can clearly differentiate between sequencing reads and inserts/fragments that would be helpful.

Thank you for the comment. Fig. 1d was simplified and detailed explanations were added to make them easier to understand.

- Vertical green bars were deleted, and all mismatched/mutated bases were represented by red or blue vertical bars.
- Dark-red, green, or light-blue horizontal bars represents sequences of other distant regions of the genome.
- A short text, “Sequences from distant homologous regions”, is added to Fig. 1d.
- We clarified that each horizontal bar indicates reference genomic sequences or mapped reads by adding short texts in the Fig. 1d.

Fig. 1d was modified.

d. The MicroSEC algorithm is based on three criteria. Filter 1, 3: the distance from the mutation position to the most distant mapped base is distributed over a probabilistically improbable limited range for any reads. Filter 2: MICR-originated sequencing errors are generated when two palindromic sequences are in the same DNA fragment. Filter 4: The mis-annealing of ssDNA derived from other distant homologous regions of the genome also creates chimeric reads and artifacts. Dark-red, green, or light-blue horizontal bars represent sequences of other distant regions of the genome. Chimeric reads with mutated bases were formed.

3. Does Fig. 1e represent a single sample or the aggregate of multiple samples?

Thank you for the comment. Fig. 3 (Fig. 1e in the previous version) shows the combined results of the validation study by amplicon-based sequencing of mutations detected in multiple samples. Due to the small number of validated mutations in the previous version, we added other 67 unique mutations derived from 31 samples. With the total of 97 mutations, the sensitivity and specificity of MicroSEC in the validation study were 97% (95% confidence interval (CI): 82%–100%) and 96% (95% CI: 88%–99%), respectively. Of the four mutations with discordant results, two (*BRCA2* p.Arg2520* and *ESR1* p.Arg394Cys) were CG-to-TG mutations. *NFYA* p.Gln155Pro (VAF 5.4% by target sequencing) was filtered out by MicroSEC but detected by amplicon-based sequencing at a low frequency of 1.6%, so it could be a true mutation. *CENPA* p.Leu91Pro, which was recognized as a true mutation by MicroSEC, was not detected by amplicon-based sequencing and may be an mapping error with low-quality bases. We added Fig. S8 to explain the discordant results between MicroSEC and amplicon-based sequencing.

Fig. 3 and Fig. S8 were added.

		Amplicon-based sequencing		
		True mutations	Artifacts	Total
MicroSEC Prediction	True mutations	28	3	31
	Artifacts	1	65	66
	Total	29	68	97

Figure 3.

Figure 3. Mutation validation by amplicon-based sequencing.

The mutations that passed through the MicroSEC filter were detected with a similar level of variant allele frequency (VAF) by both capture-based sequencing and amplicon-based sequencing (blue), with the

exception of a *CENPA* mutation. Of the five potential CG-to-TG artifacts (red), two mutations in *ESR1* or *BRCA2* were not amplified by amplicon-based sequencing. Filtered out mutations were not detected by amplicon-based sequencing (green), with the exception of a *NFYA* mutation.

a

NFYA p.Gln155Pro

b

CENPA p.Leu91Pro

Supplementary Figure 8. Aligned reads in capture-based sequencing visualized by Integrative Genomics Viewer.

a. AG-to-CT mutation in the *NFYA* gene is shown. All reads with mutations have a short supporting length from the mutated base to the end of the read (red line). **b.** T-to-C mutation in the *CENPA* gene is shown. Of the 954 reads mapped to the mutated base, 227 reads (24%) were of low quality and failed to call bases, 689 were wild-type (T), 47 were C, and one was A. Low quality bases are indicated by N. The mate-read of the green colored read is mapped to a different chromosome.

The following sentences were added.

Lines 206–220: With a total of 97 mutations, the sensitivity and specificity of MicroSEC in the validation study were 97% (95% confidence interval (CI): 82%–100%) and 96% (95% CI: 88%–99%), respectively. Of the four mutations with discordant results, two were CG-to-TG mutations (*BRCA2* p.Arg2520* and *ESR1* p.Arg394Cys). These artifacts arose from deaminated cytosines. The Q5 DNA polymerase used in the amplicon-based sequencing could not amplify such degenerated template DNA. *NFYA* p.Gln155Pro (VAF 5.4% by target sequencing) was filtered out by MicroSEC but detected by amplicon-based sequencing at a low frequency of 1.6%. The aligned reads in capture-based sequencing visualized by Integrative Genomics Viewer suggested that the mutation was a hairpin-derived artifact. However, it could be a true mutation (Supplementary Fig. 8a). *CENPA* p.Leu91Pro, recognized as a true mutation by MicroSEC, was not detected by amplicon-based sequencing. Of the 954 reads mapped to the mutated base in capture-based sequencing, 227 (24%) were of low quality and failed to call bases, 689 were wild-type (T), 47 were C, and one was A (Supplementary Fig. 8b). We considered that the mutation could be an artifact caused by miscalls due to low quality reads.

Lines 317–319: If the VAF of a mutation by amplicon-based sequencing was less than 30% of the VAF of the mutation by capture-based sequencing, we determined that the mutation was an artifact.

Lines 417–420: **Statistics** Statistical analyses were performed with R version 4.0.5. The local regression curve in Supplementary Fig. 3 was drawn by ggplot2 package version 3.3.3. Sensitivity and specificity of MicroSEC was estimated with epiR package version 2.0.19.

4. The authors note that both MuTect and VarScan were used for variant calling. Was the intersection or union of variants used?

Thank you for the comment. Since a single algorithm may fail to detect important mutations, the union (sum set) of mutations detected in MuTect and VarScan was used for the analysis.

The following sentence was added.

Lines 324–326: Since a single algorithm may fail to detect important mutations, the union of somatic mutations identified using MuTect2 (GATK v4.1.3.0)²³, VarScan2 (v2.4.3)¹⁶, and our in-house pipeline was used for the analysis.

5. In the manuscript, targeted capture sequencing is employed. How many genes or sites were analyzed per sample (using the OncoPanel described in Methods)? What was the approximate length in base pairs of each captured region? A brief description of the OncoPanel analyzed would be helpful.

Thank you for the comment. The capture probes for the Today OncoPanel were designed to examine 478 cancer-related genes, and the panel including 15,600 capture probes. As the total size of target regions was 3.4 megabases, the average length of captured regions was approximately 220 base pairs.

The following sentences were added.

Lines 133–137: We examined the sensitivity of MicroSEC in distinguishing true mutations from FFPE artifacts with our custom-made multi-gene panel test, “Today OncoPanel”. The panel including 15,600 capture probes were designed to examine 478 cancer-related genes. As the total size of target regions was 3.4 megabases, the average length of captured regions was approximately 220 base pairs.

6. In Supplementary Fig. 2a, variants from normal breast tissues were analyzed. Was this paired breast vs. blood analysis to detect somatic variants in the normal breast tissue? Please also indicate the number of mutations represented by each panel in Supplementary Fig. 2.

Thank you for the comment. We analyzed only somatic mutations in Fig. S2, as germline mutations detected in paired blood samples were excluded. The total number of mutations were added in each panel.

Fig. S2 was corrected.

a**b**
Supplementary Figure 2. The distribution of the variant allele frequencies of the breast tissue samples.

a. FFPE samples of normal breast tissues ($n = 190$) with total somatic mutations (upper), mutations filtered out by MicroSEC filter (middle), and mutations passing through the filter (lower). **b.** FFPE samples of breast tumor tissues ($n = 33$) with total somatic mutations (upper), mutations filtered out by MicroSEC

filter (middle), and mutations passed through the filter (lower). FFPE, formalin-fixed and paraffin-embedded.

7. It would be helpful if the authors could show an application of MicroSEC on FFPE exome or FFPE WGS data, which would greatly expand the utility of the tool.

Thank you for the comment. We applied MicroSEC on whole exome sequencing of the 14 matched frozen and FFPE primary cancer samples (12 colorectal adenocarcinoma, two oral squamous cell carcinoma). Our mutation caller identified 107.0 and 118.2 mutations per sample in the FF and FFPE samples, respectively. Only 0.6 mutations per sample (0.5%) detected in the FF samples were filtered by the filter, while 10.3 mutations per sample (8.7%) were filtered out in the FFPE samples. This result suggests that ssDNA-derived artifacts are also present in the sequencing data of frozen samples. With sufficient coverage, MicroSEC is applicable for whole exome sequencing data.

Fig. S4b and Table S3 were added.

a

b

Whole exome sequencing on matched samples of primary cancer

Supplementary Figure 4. The mutations detected in matching FF and FFPE samples.

a. The somatic mutations detected in the target sequencing of matching sets of frozen (blue) and FFPE (peach) breast cancer samples from eight patients. Eleven mutations found in only FFPE samples were filtered by MicroSEC (green). All mutations found in fresh frozen samples passed through the MicroSEC filter. **b.** The somatic mutations detected in the whole exome sequencing of matching sets of frozen (blue) and FFPE (peach) primary cancer samples from 14 patients. The 123 mutations (21.0%) found in only FFPE samples were filtered by MicroSEC (green). Eight mutations (0.5%) found in fresh frozen samples were filtered by the MicroSEC filter. FFPE, formalin-fixed and paraffin-embedded.

	Matched primary cancer samples	
	Fresh frozen (N = 14)	FFPE (N = 14)
Total reads (in millions)	111.8 (45.2–145.9)	142.7 (83.6–235.4)
Mapped reads (%)	93.3 (92.8–93.7)	93.4 (85.2–94.1)
Unique reads (%)	86.3 (83.8–93.0)	86.5 (73.5–92.2)
Mean coverage	199 (83–261)	255 (134–394)
Median insert size (base)	223 (197–238)	173 (124–205)
Somatic mutations removed by		
Filter 1	0.1 (0–1)	8.2 (0–47)
Filter 2	0 (0–0)	3.9 (0–23)
Filter 3	0.1 (0–1)	7.3 (0–42)
Filter 4	0.4 (0–3)	1.2 (0–4)
Any of Filter 1–4	0.6 (0–3)	10.3 (0–55)
Mutations passing the filter	106.4 (81–196)	107.9 (85–138)
Filtered rate (%)	0.5	8.7
CG-to-TG potential artifacts	NA	45.8 (14–56)
Intra \geq 10-base homopolymer	0.0 (0–0)	0 (0–0)
Remaining mutations	106.4 (81–196)	62.1 (45–89)

Data are shown as mean (range).
NA, not applicable; FFPE, formalin-fixed and paraffin-embedded.

Supplementary Table 3. MicroSEC filtering summary for whole exome sequencing.

The following sentences were added.

Lines 269–272: The study cohort comprised 26 patients with breast cancer who underwent tumor resection or prophylactic surgery because of hereditary breast and ovarian cancer syndrome at St. Luke’s International Hospital between 2010 and 2020, and 14 patients with primary cancer at National Cancer Center Hospital.

Lines 277–280: We also obtained 14 matched FF and FFPE primary cancer samples from 12 colorectal adenocarcinoma patients and two oral squamous cell carcinoma patients. The surgeries were performed between 2012 and 2019. Tissue samples were provided by the National Cancer Center Biobank, Japan.

Lines 284–286: The study protocol was approved by the Ethics Committees of St Luke's International Hospital, National Cancer Center Hospital, and the National Cancer Center Research Institute.

Lines 287–289: **Capture-based panel sequencing and whole exome sequencing**

Breast samples were subjected to capture-based panel sequencing, and primary cancer samples were subjected to whole exome sequencing.

Lines 292–299: Lastly, 500 ng or 50 ng of genomic DNA was subjected to target fragment enrichment using SureSelectXT Custom kits (Agilent Technologies, USA) or Twist Custom Panels (Twist Bioscience, USA) for capture-based panel sequencing, respectively. Libraries for whole exome sequencing were generated from genomic DNA (50 ng) using Twist Library Preparation EF kit and Twist Universal adaptor system (Twist Bioscience). Enrichment of exonic fragments using Twist Human Comprehensive Exome Panel kit and Twist Fast Hybridization and wash kit (Twist Bioscience).

Lines 440–441: National Cancer Center Biobank is supported by the National Cancer Center Research and Development Fund, Japan.

8. The authors look for palindromes “within 150 bases of the neighboring sequence.” What was the insert length used by the authors for capture sequencing? Relatedly, does MicroSEC look for palindromes/homology within sequencing inserts, individual reads, or both?

Thank you for the comment. The median insert size of FFPE normal breast tissue samples for target sequencing was 158 (Table 1), and we realized that 150 bases may be too narrow a range to search for palindromes. We counted palindromes in FFPE normal breast samples and frozen breast tumor samples with three different ranges to search for palindromes: 150 bases, 200 bases, and 300 bases (Fig. S7). No palindromes were detected in frozen tumor samples. When the search range was 150 bases, 430 palindromes were detected, and when the search range was extended to 200 bases, 438 palindromes were detected; extending the search range to 300 bases did not increase the number of palindromes. Based on these results, we concluded that the search range of 200 bases was appropriate and reanalyzed all the data.

MicroSEC looks for palindromes/homology within each read. A mutation is determined as an artifact when a certain percentage of the reads with the mutation contain palindromes/homology. The details are shown in Fig. 2.

Fig. S6a was added.

a**b****c****d**
FFPE normal breast tissue

Frozen breast tumor

Supplementary Figure 6. The optimal hyperparameters of MicroSEC.

Detected artifacts with various hyperparameters in 190 FFPE normal breast tissue (gray) and 23 frozen breast tumor (black) samples. The base length to search palindromes (a), P-value thresholds for Filters 1 and 3 (b), Filter 2 (c), and Filter 4 (d) were varied and the number of artifacts detected was counted. FFPE, formalin-fixed and paraffin-embedded.

The following sentences were added.

Lines 175–187: We further validated the filtering hyperparameters and thresholds. The median insert size of FFPE normal breast tissue samples for target sequencing was 158 (Table 1). We counted palindromes in FFPE normal breast samples and frozen breast tumor samples with three different ranges (150, 200, and 300 bases) to search for palindromes: (Supplementary Fig. 6a). No palindromes were detected in frozen tumor samples. Using the search range of 150 bases, 430 palindromes were detected. When the search range was extended to 200 bases, we could detect 438 palindromes. Interestingly, extending the search range to 300 bases did not increase the number of detected palindromes. Based on these results, we concluded that the search range of 200 bases was appropriate. However, all analyses using any thresholds for filter 1–4 did not filter out mutations detected in frozen tumor samples and we could not identify the optimal thresholds. Further analyses with more tumor-derived mutations would thus be necessary to determine the optimal thresholds.

9. The authors state “ $p < 10^{-6}$ are considered artifacts”. How was this cutoff determined?

Thank you for the comment. This answer is a duplicate of major comment (b).

To evaluate the relationship between mean coverage and artifact detection rate, we performed random sampling from sequences of FFPE samples of normal breast tissue and frozen samples of breast cancer. From each library, three random sampling runs were performed so that the mean coverage of each library was 400, 300, 200, 100, 50, and 30. We calculated the average artifact detection rate after excluding mutations for which there were no reads calling. We also varied the p-value threshold for Filter 1 and Filter 3 from 10^{-2} to 10^{-9} to determine the appropriate threshold. The results are shown in Fig. S7. The results showed that the detection rate of artifacts decreased, and the number of false positives increased when the average depth of coverage was less than 100. In addition, false positives were seen in frozen tumor samples when the p-value threshold was above 10^{-5} , indicating that the threshold should be below or equal to 10^{-5} . The threshold can be 10^{-5} , but to avoid over-filtering we adopted 10^{-6} .

Fig. S6b and Fig. S7 were added.

Supplementary Figure 6. The optimal hyperparameters of MicroSEC.

Detected artifacts with various hyperparameters in 190 FFPE normal breast tissue (gray) and 23 frozen breast tumor (black) samples. The base length to search palindromes (a), P-value thresholds for Filters 1 and 3 (b), Filter 2 (c), and Filter 4 (d) were varied and the number of artifacts detected was counted. FFPE, formalin-fixed and paraffin-embedded.

Supplementary Figure 7. Appropriate mean coverage and P-value for MicroSEC.

The relationship between mean coverage, threshold of P-value, and artifact detection rate in FFPE normal breast tissue (a) and frozen breast tumor (b) are shown. From each library, three random sampling runs were performed so that the mean coverage of each library was 400, 300, 200, 100, 50, and 30. The average artifact detection rates (lines) and called mutation counts were calculated. The P-value thresholds for

Filters 1 3 were varied from 10^{-2} to 10^{-9} to determine the appropriate threshold. The artifact detection rate decreased, and the number of false positives increased when the average depth of coverage was less than 100. In addition, false positives were seen in frozen tumor samples when the P-value threshold was above 10^{-5} . FFPE, formalin-fixed and paraffin-embedded.

The following sentences were added.

Lines 188–198: To evaluate the relationship between the mean coverage and artifact detection rate, we performed random sampling from the sequences of FFPE normal breast tissue samples and frozen breast cancer samples. From each library, three random sampling runs were performed so that the mean coverage of each library was 400, 300, 200, 100, 50, and 30. We calculated the average artifact detection rate after excluding mutations for which there were no reads observed. We also varied the P-value threshold for Filters 1 and 3 from 10^{-2} to 10^{-9} to determine the appropriate threshold (Supplementary Fig. 7). The artifact detection rate decreased, and the number of false positives increased when the average depth of coverage was less than 100. In addition, false positives were seen in frozen tumor samples when the P-value threshold was above 10^{-5} , indicating that the threshold should be below or equal to 10^{-5} . Although the threshold could be 10^{-5} , we adopted 10^{-6} to avoid overfiltering.

10. The authors state “VarScan2 mutation caller employs it as a filtering.” Could the authors clarify what “it” refers to?

Thank you for the comment. The text has been revised as follows to clearly convey the meaning.

The following sentence was added.

Lines 102–104: The VarScan2 mutation caller empirically employs whether the mutated bases are biased toward the ends of the reads as a filtering metric, but it does not take soft-clipping into account(1).

11. Does MicroSEC act on BAM files? Is a certain type of alignment required? A brief (1-2 sentence) technical explanation would be helpful in the manuscript text.

Thank you for the comment. Since MicroSEC analysis is based on the phenomenon that reads supporting mutations are statistically mapped in a biased manner, FASTQ files with only sequence information are insufficient, and BAM files with alignment information are necessary. In addition, lists of mutation-supporting read IDs, which were required in past versions, are no longer required in the current version. Along with the major comment #a, the following text has been added to the main text.

The following sentences were added.

Lines 109–115: Based on our theory that artifacts are derived from ssDNA-annealing, we have developed a MICR-originating Sequence Error Cleaning pipeline (MicroSEC), a post hoc filtering pipeline to predict if a given mutation is an MICR-derived error. This pipeline allows the processing of thousands of mutations of target sequencing data within hours on a standard PC with 16 gigabytes of memory. MicroSEC requires a list of mutations and corresponding BAM files, rather than FASTQ files as it uses the positional bias of reads mapped against mutations.

12. In Supplementary Fig. 4, the authors show 56 FFPE-specific variants which were not filtered out by MicroSEC. What would be the plausible explanation for these artifacts? Could the authors suggest approaches that might filter these variants?

Thank you for the comment. In Fig. S4, DNA was extracted from frozen specimens about 3 mm in size and from thinly sliced FFPE specimens. It was necessary to elaborate that the matched specimens were not collected from completely the same sites. Also, tumors had heterogeneity, and the mutations harbored in each site were different. Since frozen specimens consisted of many subclones, only mutations common to many subclones were detected as VAF >5%, whereas FFPE specimens comprised only a small number of subclones, and therefore subclone-specific mutations were also detected. This was the reason why 56 unfiltered mutations were detected in the FFPE samples, and we consider these to be true mutations.

The following sentences were added.

Lines 161–172: Notably, the MicroSEC did not remove any mutations detected in matched FF samples (Supplementary Fig. 4a). Eight mutations were detected in the frozen samples only, and 42 mutations in both the frozen and FFPE samples, all of which passed through the MicroSEC filter. Sixty-five mutations were detected in FFPE samples only, of which 11 were detected as artifacts. It was necessary to elaborate that the matched samples were not collected from the exact same sites. DNA was extracted from frozen specimens of approximately 3 mm and thinly sliced FFPE specimens. The mutations found in each site were different due to tumor heterogeneity. Since frozen specimens consisted of multiple subclones, only common mutations were detected with VAF >5%, whereas FFPE specimens comprised only a small number of subclones, subclone-specific mutations were thus also detected. This was the reason why 56 unfiltered mutations were detected in the FFPE samples, which we consider to be true mutations.

Reviewer #2 (Remarks to the Author):

The main result of this paper is that using the filtering algorithm pipeline (MicroSEC) that they developed could potentially filter formalin-fixed and paraffin-embedded (FFPE) mutational artefacts from post hoc DNA sequencing. The lack of availability of fresh or fresh frozen (FF) samples leads to the use of FFPE

samples for sequencing purposes, resulting in a high number of variants calling errors. FFPE variant calling errors are the result of material degradation and are characterized by a high number of single-stranded DNA (ssDNA). The paper also claims that microhomology induced chimeric read (MICR) due to ssDNA is the major cause of error-prone sequencing aside from the well-known CG-to-TG mutational artefacts patterns in FFPE.

In their paper, the authors have tested their theory, performing their filtering pipeline approach in breast cancer and normal breast tissue with some independent and some matching FF and FFPE samples. A weak but understandable point of the filtering usage is the requirement of at least 400-reads mean coverage. Since the authors used their method in target sequencing data, the requirement could be met, but this would not be as easy to accomplish in common whole-genome sequencing. However, the approach could potentially be useful in gene panel sequencing as they demonstrated. As could be expected from a filtering tool, the paper method points out two important facts. First is that mutations which passed through the filter tended to have a higher variant allele frequency (VAF), and second, that the MicroSEC did not remove any mutation detected in matched FF samples. Both are expected in a well-working tool for filtering artefacts and selecting those real mutations.

I found the method reliable, making use of many thoughtful and deliberated steps to account for many different possible scenarios for filtering artefacts; here are some considerations to add to my perspective:

1.- The most important considerations I see is that the paper claims there are some methods already available for artefact filtering. However, they are “not highly accurate” (line 148). The question is, what is the basis of this claim? No reference is given here, and most importantly, no comparison is made between the performance of their method and those of the “not highly accurate” algorithms on uniform data, mutations, and filtering artefacts. This should be taken into account.

Thank you for the comment. For example, Strand Orientation Bias Detector (SOBDetector), the most recent filtering algorithm for FFPE sequencing artifacts, was applied to sequence data of frozen breast tumor samples and FFPE normal breast tissue samples (Figure X). Based on the assumption that formalin modification occurs only in one of the strands, SOBDetector detects artifacts based on the strand-bias of the detected mutations, and the creators claim that it shows a state-of-the-art performance that can predict artifacts with 90% accuracy. Although SOBDetector identified all 74 single nucleotide variants (SNVs) detected in frozen breast tumor samples as true mutations, it was only able to detect three of the 1,351 SNVs detected in FFPE normal breast tissue samples as artifacts (SOBDetector is an algorithm that can only be applied to SNVs). We think that this algorithm is not highly accurate.

Fig. S9 was added.

a

b

Supplementary Figure 9. Filtering efficiency of SOBDetector and MicroSEC.

a. The single nucleotide variants detected in the frozen breast tumor samples were subjected to SOBDetector and MicroSEC. All 74 variants passed through the both filters. **b.** The single nucleotide variants detected in the FFPE normal breast tissue samples were subjected to SOBDetector and MicroSEC.

Of the 1,351 variants, only three variants were filtered out by SOBDetector, whereas MicroSEC detected 1,199 variants as possible artifacts. FFPE, formalin-fixed and paraffin-embedded.

The following sentences were added.

Lines 243–252: For example, Strand Orientation Bias Detector (SOBDetector)(2), the most recent filtering algorithm for FFPE sequencing artifacts, was applied to sequence data of frozen breast tumor samples and FFPE normal breast tissue samples (Supplementary Fig. 9). Based on the assumption that formalin modification occurs only in one of the strands, SOBDetector detects artifacts based on the strand bias of the detected mutations, and the creators claim that it shows a state-of-the-art performance that can predict artifacts with 90% accuracy. Both SOBDetector and MicroSEC identified all 74 single nucleotide variants (SNVs) detected in frozen breast tumor samples as true mutations. SOBDetector was able to detect only three of the 1,351 SNVs detected in FFPE normal breast tissue samples as artifacts, whereas MicroSEC detected 1,199 SNVs as artifacts.

Lines 414–416: **SOBDetector**

SOBDetector (v1.0.2) is applied to 74 SNVs detected in frozen breast tumor samples and 1,351 SNVs detected in FFPE normal breast tissue samples with default settings.

2.- Predictions were made based on 11 mutations that passed through the MicroSEC filtering, and 14 were not detected. The authors should elaborate more fully as to whether 11 detected and 14 non-detected mutations are reliable enough results to determine the success of their method. Are they enough?

Thank you for the comment. As pointed out, the validation study with 11 true mutations, 14 artifacts, and five CG>TG mutations is not enough size, so we added other 67 unique mutations derived from 31 samples. With the total of 97 mutations, the sensitivity and specificity of MicroSEC in the validation study were 97% (95% confidence interval (CI): 82%–100%) and 96% (95% CI: 88%–99%), respectively. Of the four mutations with discordant results, two (BRCA2 p.Arg2520* and ESR1 p.Arg394Cys) were CG-to-TG mutations. NFYA p.Gln155Pro (VAF 5.4% by target sequencing) was filtered out by MicroSEC but detected by amplicon-based sequencing at a low frequency of 1.6%, so it could be a true mutation. CENPA p.Leu91Pro, which was recognized as a true mutation by MicroSEC, was not detected by amplicon-based sequencing and may be a mapping error with low-quality bases. We added Fig. S8 to explain the discordant results between MicroSEC and amplicon-based sequencing.

Fig. 3 and Fig. S8 were added.

		Amplicon-based sequencing		
		True mutations	Artifacts	Total
MicroSEC Prediction	True mutations	28	3	31
	Artifacts	1	65	66
	Total	29	68	97

Figure 3.

Figure 3. Mutation validation by amplicon-based sequencing.

The mutations that passed through the MicroSEC filter were detected with a similar level of variant allele frequency (VAF) by both capture-based sequencing and amplicon-based sequencing (blue), with the

exception of a *CENPA* mutation. Of the five potential CG-to-TG artifacts (red), two mutations in *ESR1* or *BRCA2* were not amplified by amplicon-based sequencing. Filtered out mutations were not detected by amplicon-based sequencing (green), with the exception of a *NFYA* mutation.

a

b

Supplementary Figure 8. Aligned reads in capture-based sequencing visualized by Integrative Genomics Viewer.

a. AG-to-CT mutation in the *NFYA* gene is shown. All reads with mutations have a short supporting length from the mutated base to the end of the read (red line). **b.** T-to-C mutation in the *CENPA* gene is shown. Of the 954 reads mapped to the mutated base, 227 reads (24%) were of low quality and failed to call bases, 689 were wild-type (T), 47 were C, and one was A. Low quality bases are indicated by N. The mate-read of the green colored read is mapped to a different chromosome.

The following sentences were added.

Lines 206–220: With a total of 97 mutations, the sensitivity and specificity of MicroSEC in the validation study were 97% (95% confidence interval (CI): 82%–100%) and 96% (95% CI: 88%–99%), respectively. Of the four mutations with discordant results, two were CG-to-TG mutations (*BRCA2* p.Arg2520* and *ESR1* p.Arg394Cys). These artifacts arose from deaminated cytosines. The Q5 DNA polymerase used in the amplicon-based sequencing could not amplify such degenerated template DNA. *NFYA* p.Gln155Pro (VAF 5.4% by target sequencing) was filtered out by MicroSEC but detected by amplicon-based sequencing at a low frequency of 1.6%. The aligned reads in capture-based sequencing visualized by Integrative Genomics Viewer suggested that the mutation was a hairpin-derived artifact. However, it could be a true mutation (Supplementary Fig. 8a). *CENPA* p.Leu91Pro, recognized as a true mutation by MicroSEC, was not detected by amplicon-based sequencing. Of the 954 reads mapped to the mutated base in capture-based sequencing, 227 (24%) were of low quality and failed to call bases, 689 were wild-type (T), 47 were C, and one was A (Supplementary Fig. 8b). We considered that the mutation could be an artifact caused by miscalls due to low quality reads.

Lines 317–319: If the VAF of a mutation by amplicon-based sequencing was less than 30% of the VAF of the mutation by capture-based sequencing, we determined that the mutation was an artifact.

Lines 417–420: **Statistics** Statistical analyses were performed with R version 4.0.5. The local regression curve in Supplementary Fig. 3 was drawn by ggplot2 package version 3.3.3. Sensitivity and specificity of MicroSEC was estimated with epiR package version 2.0.19.

3.- Regarding the MicroSEC pipeline: it's not clear, and I don't find any reasonable excuse for performing the adapter sequence trimming at the second step instead of doing it during the first step as part of a normal QC process. And once the reads pass the QC steps, the pipeline could start with the position of the mutated bases that have been detected. If there is a reason for this ordering, the authors should explain it, as it is not clear.

Thank you for the comment. As pointed out, adapter trimming is usually performed in the quality check step. Especially in de novo sequencing, adapter trimming is a must. However, when the reference sequence already exists and the reads are mapped, adapter trimming is not always performed because the mapping algorithm will soft-clip the adapter sequences. We had added an adaptor trimming step to MicroSEC to

allow post hoc analysis of data from facilities that do not routinely perform adaptor trimming in their analyses, including ours.

The following sentence was added.

Lines 345–348: Second, the adapter sequence at the 3' end of each read was trimmed with the trimLRPatterns function from the Biostrings package with the maximum mismatch rate of 10%, so as to analyse libraries that did not perform adaptor trimming in the quality check step.

4. I also had some concerns about some of the figures. Starting from Supplementary Figure 2. As I understand, Top 2a and Top 2b represent the total number of mutations detected, with their corresponding VAF. How is it possible that the y-axis figure a goes up to 900 mutations, with the highest bar not reaching 1000, while in the “Filtered out mutations (normal, FFPE)” (which suggests that it should include fewer mutations) is also nearly 1000 mutations? Also, in “Mutations passing the filter (normal, FFPE),” the max bar is over 150 mutations. The sum of “Filtered out mutations (normal, FFPE)” and “Mutations passing the filter (normal, FFPE)” should be equal to “Total mutations (normal, FFPE),” and clearly the sum is over that. In the figure b, the numbers seem reasonable, but not in a. Supplementary Figure 4. Same question as in the point 2. Are those numbers enough to determine the successful of the method? Supplementary Figure 5. Regarding the appropriate thresholds confirmed in this figure: I can see in Figure 5a that the threshold could be appropriated for filtering purposes, but certainly in 5c and 5d, that threshold in 50% and 15% respectively could be improved. And more explanation about why are appropriated could be helpful.

Thank you for the comment. We corrected ambiguous points in the figures to make them clearer.

- Regarding Fig. S2, the histogram itself was correct, but the labels of Y-axes were inappropriate. We re-labeled the Y-axes.

- The total number of mutations were added in each panel of Fig. S2 and Fig. S5.

- In the previous version of the paper, eight matched frozen and FFPE samples of breast cancer were compared for the performance evaluation of MicroSEC, but the number of cases was not sufficient. We added a comparison of WES analysis of matched frozen and FFPE samples from 14 primary cancer patients. The results are shown in Fig. S4b.

- We added Fig. S6 concerning the filtering rates for various changes in the MicroSEC thresholds. All analyses using any of the thresholds did not filter out mutations detected in frozen tumor samples, but the number of mutations was not large enough to identify the optimal thresholds. Further analyses with more tumor-derived mutations are needed to determine the optimal thresholds.

Fig. S2 and Fig. S5 were corrected. Fig. S4b and Fig. S6 were added.

a

b

Supplementary Figure 2. The distribution of the variant allele frequencies of the breast tissue samples.

a. FFPE samples of normal breast tissues (n = 190) with total somatic mutations (upper), mutations filtered out by MicroSEC filter (middle), and mutations passing through the filter (lower).

b. FFPE samples of breast tumor tissues (n = 33) with total somatic mutations (upper), mutations filtered out by MicroSEC filter (middle), and mutations passed through the filter (lower).

FFPE, formalin-fixed and paraffin-embedded.

Supplementary Figure 5. The distribution of the mutations in breast tissue samples.

a. The rate of low-quality bases in mutation-supporting reads in 190 FFPE normal breast tissue samples (left) and 23 frozen breast tumor samples (right). **b.** The rate of soft-clipped reads in FFPE samples of normal breast tissue (left) and frozen samples of breast tumor (right). **c.** The rate of reads derived from other homologous regions in FFPE samples of normal breast tissue (left) and frozen samples of breast tumor (right). **d.** The rate of reads derived from the hairpin structure in FFPE samples of normal breast tissue (left) and frozen samples of breast tumor (right). Dotted red lines represent the thresholds.

a

b

Whole exome sequencing on matched samples of primary cancer

Supplementary Figure 4. The mutations detected in matching FF and FFPE samples.

a. The somatic mutations detected in the target sequencing of matching sets of frozen (blue) and FFPE (peach) breast cancer samples from eight patients. Eleven mutations found in only FFPE samples were filtered by MicroSEC (green). All mutations found in fresh frozen samples passed through the MicroSEC filter. **b.** The somatic mutations detected in the whole exome sequencing of matching sets of frozen (blue) and FFPE (peach) primary cancer samples from 14 patients. The 123 mutations (21.0%) found in only FFPE samples were filtered by MicroSEC (green). Eight mutations (0.5%) found in fresh frozen samples were filtered by the MicroSEC filter. FFPE, formalin-fixed and paraffin-embedded.

Supplementary Figure 6. The optimal hyperparameters of MicroSEC.

Detected artifacts with various hyperparameters in 190 FFPE normal breast tissue (gray) and 23 frozen breast tumor (black) samples. The base length to search palindromes (**a**), P-value thresholds for Filters 1 and 3 (**b**), Filter 2 (**c**), and Filter 4 (**d**) were varied and the number of artifacts detected was counted. FFPE, formalin-fixed and paraffin-embedded.

The following sentences were added.

Lines 175–187: We further validated the filtering hyperparameters and thresholds. The median insert size of FFPE normal breast tissue samples for target sequencing was 158 (Table 1). We counted palindromes in FFPE normal breast samples and frozen breast tumor samples with three different ranges (150, 200, and 300 bases) to search for palindromes: (Supplementary Fig. 6a). No palindromes were detected in frozen tumor samples. Using the search range of 150 bases, 430 palindromes were detected. When the search range was extended to 200 bases, we could detect 438 palindromes. Interestingly, extending the search range to 300 bases did not increase the number of detected palindromes. Based on these results, we concluded that the search range of 200 bases was appropriate. However, all analyses using any thresholds for filter 1–4 did not filter out mutations detected in frozen tumor samples and we could not identify the optimal thresholds. Further analyses with more tumor-derived mutations would thus be necessary to determine the optimal thresholds.

5. Lastly, there is a lack of clarity starting from line 251 in the Supporting length analysis that makes it difficult to understand the method. I can't give any strong opinion if the mathematical method is robust as I don't get the meaning of it. I became completely lost beginning with “no more than $N+1$ matching sequences outside the gap”, which I think should be clarified. The same thing happens on line 150, causing things to be unclear.

Thank you for the comment. In order for the mapper to detect the mutation near the ends, it must satisfy a condition that the mutation is not soft-clipped in each read. In the previous version of the paper, the condition was expressed only in mathematical equations. To make it easier to understand, we added Fig S10a. In addition, when an indel mutation occurs in a repetitive sequence, the reads must support beyond the end of the repeating sequence in order to call the mutation. We added Fig. S10b to show the detail.

Fig. S10 was added.

a

Reference sequence

5' CGAGCACTGTGTCAGGCTGTGGCTGAGCCCCAAGGCCCAAACATGTGCC 3'

b Reference sequence

5' ATCTAGCTCGAGCACTCAACAACAACAACAACAACAACAACAATATGTGCC 3'

Supplementary Figure 10. Limitation on the number of bases to map around a mutation.

a. L was considered to be the read length, N the number of bases mutated, and M the number of bases mapped outside the mutation. When Burrows-Wheeler Aligner were used as a mapper, the penalty due to an N -base mutation was $N + 6$, the soft-clipping penalty was 5, and the point for mapped M bases was M . When the mutation is called and not soft-clipped, $M > N + 1$ must be satisfied regardless of the type of mutation. **b.** If the number of repetitions changes in a short tandem repeat, only reads containing all the repetitive sequences can support the presence of indel mutations.

The following sentences were modified.

Lines 368–375: Furthermore, we need to consider the penalties by Burrows-Wheeler Aligner for mapping. Since the penalty due to an N -base mutation is $N + 6$, the soft-clipping penalty is 5, and the point for mapped M bases is M , a sequence supporting the mutation would be soft-clipped if there are no more than $N + 1$ matching sequences outside the gap (Supplementary Fig. 10a). Given the 5' and 3' repetitive sequences around the mutation of lengths $R_{5'}$ and $R_{3'}$, the supporting lengths can be distributed between $\max(R_{3' \text{ or } 5'}, N + 1)$ and $L - \max(R_{5' \text{ or } 3'}, N + 1)$ for deletions, or between $\max(R_{3' \text{ or } 5'}, N + 1)$ and $L - N - \max(R_{5' \text{ or } 3'}, N + 1)$ for insertions, with $\max(X, Y)$ denoting the larger value of X and Y (Supplementary Fig. 10b).

Overall, the idea and goal are clear, and steps are thoughtfully selected. The method could potentially influence certain future sequencing perspective as it paves a way to reanalyse older data or new FFPE samples which were never used due to artefacts problems. However, the paper needs further comparison with available tools to demonstrate exactly what their improvement is.

Reviewers' comments:

Reviewer #1 (Remarks to the Author):

The authors have effectively addressed many concerns raised in the previous review. However, several minor points remain unclarified plus a few additional concerns related to the new content.

Previous concerns

a) Previous major point: The authors have resolved the memory issues encountered when testing the previous version of MicroSEC, based on analysis of one FFPE-derived breast cancer exome sample. However, there are two additional issues with the updated version of MicroSEC. The first is that the input bam file apparently must have chromosomes indicated as "chr1" rather than "1", which led to errors when running bam files with the latter format. Please note this requirement on the Github page or make the tool able to handle chromosomes in either format, since revising bam files can be time- and space-intensive. Second, I encountered the following error after MicroSEC had apparently finished initial analysis of all mutations in my mutation list:

```
Error in str_split(df_mutation[i, "Mut_type"], "-")[[1]][[2]] :
```

```
subscript out of bounds
```

```
Calls: fun_read_check
```

```
Execution halted
```

This was apparently related to splitting the "Mut_type" field in the input .xlsx mutation file, where all of my mutations had "1-snv" entered for this field, so it is unclear why the splitting to obtain the second field did not work. If there is a way to help users avoid this issue in the Github page that would be helpful."

b) Previous minor point #1: The description of Fig. 1 and its legend are helpfully improved, but there are still points where it is unclear. For example, Fig. 1a is referenced without mentioning what type of sample (e.g. which cancer type, etc.) was sequenced and by what method. A description like the following would be helpful around lines 75-80, before delving into the Results: "To better understand the spectrum of FFPE artifacts, we performed targeted sequencing of [20 breast cancer samples] using a [400-gene panel] and reviewed likely artifacts. For example, in one sample we observed an artifact in FGFR4... (Fig. 1a)." Further, in Fig. 1b, the blue and green arrows represent the "upstream" and "downstream" portions of the same read. If that is the case, why is there overlap between the two arrows even in Fig. 1b, top (unmapped read)?

c) Previous minor point #3: Please indicate the number of samples included in Fig. 3 (formerly Fig. 1e) and the type of tissue in the legend. Please also state why/how the 97 specific mutations were selected. Additionally, do the VAFs shown represent the mean or median of all samples analyzed? Some of this information is available from the point-by-point response but please make it clear for readers of the manuscript itself.

d) Previous minor point #6: Thank you for adding the number of mutations to Supplementary Fig. 2. Please also state in the legend that the somatic mutations shown represent those present in normal breast tissue but not in normal blood. Please also state in the Results section the rationale for why normal breast tissue was compared to blood to identify somatic mutations. Based on the description scattered in various section of the manuscript, the authors first tested the algorithm in normal breast tissue so that they can test the efficacy (sensitivity) of the filtering. This was followed by a testing on specificity using 23 FF and 33 FFPE breast cancer samples (lines 154-160). An outline of study design can be of great help so that the readers can understand why the pipeline was first tested using normal breast tissues.

e) Previous minor point #12: Thank you for the clarification of the FFPE-only variants in Supplementary Fig. 4. One minor typo may have been introduced: "fur to tumor heterogeneity", where I believe the authors meant "due to tumor heterogeneity."

Additional new concerns

1. The previous manuscript lacked headers for "Introduction", "Results", etc. and these headings have been added to the current revised manuscript. However, the Introduction (e.g. lines 75-89; lines 115-131) actually contains Methods and Results material, including 2 of 3 main figures (Figs. 1 and 2) being cited and described in the Introduction. I would suggest moving all/most figure descriptions to the Results section, while the Introduction should retain background information and brief transitional material such as lines 90-98 and lines 109-115, which could help to nicely wrap up the Introduction and lead into the Results section. The algorithm design described in lines 115-131 can be put into a new Result section as "Study Design". This section perhaps also should include description on method validation approaches using the three cohorts described in Table 1.

2. Is Fig. 2a an actual read or hypothetical? From what type of sample? It would be helpful to explain the details.

3. Lines 164-172 presented the following results "Sixty-five mutations were detected in FFPE samples only, of which 11 were detected as artifacts.". The authors tried to justify that the FFPE-only variants were valid due to intratumor heterogeneity. However, the normal breast FFPE tissue filtering rate presented in Supplementary Figure 3 appears to have much higher number of mutations being filtered (5 per sample) even at the highest mutation depth bin of (90-100). It is important that the authors compare the filtering rate on the breast cancer samples with those presented in Supplementary Figure 3 to ensure that the low number of mutations being filtered in FFPE breast cancer tissue was indeed due to elevated coverage.

4. The exome FFPE analysis on lines 228-238 is very helpful; thank you for this addition. Could the authors also state the relative prevalence of CG-to-TG artifacts vs. MicroSEC-filtered (MICR-related) artifacts? This appears to be shown in Supp. Table 3 but a text description would also be helpful so that users can know approximately what percentage of their total artifact burden will be removed by MicroSEC.

5. Line 151-152: "A tendency to increase could be observed for the mutation filtering rate as the number of reads with mutations increased (Supplementary Fig 3)". This statement appears to be contradictory to the data presented in Supplementary Fig 3 and the S. Fig. 3 legend which states "The mutation filtering rate (black line) is high at low depth and more than 90% of mutations are filtered by MicroSEC pipeline when the mutation coverage is 10-22."

6. The validation described in lines 199-220: please clarify that these are the mutations detected in the normal breast tissue or the breast cancer samples.

7. Lines 239-266: the content should be restructured into Discussion.

We would like to thank the Reviewer #1 for the thoughtful comments. We have revised and modified the manuscript in accordance with the reviewer's comments and suggestions. We believe that this revised version is improved due to the clear and helpful comment from the reviewer.

Find below a point-by-point response to the concerns raised by the reviewer.

Reviewer's comments:

Reviewer #1 (Remarks to the Author):

The authors have effectively addressed many concerns raised in the previous review. However, several minor points remain unclarified plus a few additional concerns related to the new content.

Previous concerns

a) Previous major point: The authors have resolved the memory issues encountered when testing the previous version of MicroSEC, based on analysis of one FFPE-derived breast cancer exome sample. However, there are two additional issues with the updated version of MicroSEC. The first is that the input bam file apparently must have chromosomes indicated as "chr1" rather than "1", which led to errors when running bam files with the latter format. Please note this requirement on the Github page or make the tool able to handle chromosomes in either format, since revising bam files can be time- and space-intensive. Second, I encountered the following error after MicroSEC had apparently finished initial analysis of all mutations in my mutation list:

```
Error in str_split(df_mutation[i, "Mut_type"], "-")[[1]][[2]] :  
subscript out of bounds  
Calls: fun_read_check  
Execution halted
```

This was apparently related to splitting the "Mut_type" field in the input .xlsx mutation file, where all of my mutations had "1-snv" entered for this field, so it is unclear why the splitting to obtain the second field did not work. If there is a way to help users avoid this issue in the Github page that would be helpful."

Thank you for pointing out the two problems to be improved in MicroSEC.

- 1) We have modified the software so that it can handle both "chr1" and "1" format chromosomes, and added the description to Github webpage. We have confirmed that the software works with the BAM file with "1"-Y" format uploaded by CCLE (<https://trace.ncbi.nlm.nih.gov/Traces/sra/?run=SRR8618961>).
- 2) Errors related to the str_split function occurred when the mutation list excel file contained blank lines. By adding a step to remove the blank lines, the error was resolved.

The following sentence was **modified**.

Line 378: MicroSEC (**v1.2.8**) is a filtering pipeline written in R language designed to discover MICR-derived sequencing errors in FFPE samples.

b) Previous minor point #1: The description of Fig. 1 and its legend are helpfully improved, but there are still points where it is unclear. For example, Fig. 1a is referenced without mentioning what type of sample (e.g. which cancer type, etc.) was sequenced and by what method. A description like the following would be helpful around lines 75-80, before delving into the Results: “To better understand the spectrum of FFPE artifacts, we performed targeted sequencing of [20 breast cancer samples] using a [400-gene panel] and reviewed likely artifacts. For example, in one sample we observed an artifact in FGFR4... (Fig. 1a).” Further, in Fig. 1b, the blue and green arrows represent the “upstream” and “downstream” portions of the same read. If that is the case, why is there overlap between the two arrows even in Fig. 1b, top (unmapped read)?

We apologize for the many points in Figure 1 that are difficult to understand.

- The sequencing data used in Fig. 1a-c were obtained by target sequencing of DNA extracted from a FFPE sample of normal breast tissue, “SL_0002_L_FFPE_11”. This sample is of low quality and has been excluded from the analysis. We clarified that a T-to-C artifact in the FGFR4 gene found in target sequencing data of a FFPE normal breast tissue sample.

- In Fig. 1b, we did not fully explain the overlap between the upstream and downstream sequences. We have described a strange phenomenon where there is overlap between the upstream and downstream sequences, and also overlap in the sequences to which each is mapped. We also clarified that Fig. 1c explains the mechanism of Fig. 1b.

The following sentences were **modified**.

Lines 92–98: **To better understand the spectrum of FFPE artifacts, we performed target sequencing of a low-quality FFPE sample of normal breast tissue using a 478-gene panel and reviewed likely artifacts.** First, we found mapping anomalies characteristic of artifacts in FFPE samples. DNA extracted from samples was fragmented at random positions to the appropriate size before sequencing. Mutated bases are expected to be distributed evenly in the reads. However, **we observed a T-to-C artifact in FGFR4 gene with a marked bias in the position of the mutation (Fig. 1a).**

Lines 100–104: The mapping of a **representative** read with an artifact in FGFR4 was examined in detail (Fig. 1b). The upstream sequence of the read was mapped to the forward strand of the genome, and the downstream sequence was mapped to the reverse strand of the same genomic region. **Strangely, the upstream and downstream sequences overlapped, as did the genomic sequences to which each was mapped.**

Fig. 1 legend: a. The genomic sequence visualized by Integrative Genomics Viewer exhibits a T-to-C artifact in the FGFR4 gene **found in target sequencing data of a FFPE normal breast tissue sample**. In all mutation-supporting reads, only six bases downstream of the mutation were mapped, and the rest is soft-clipped (red line). The blue colored read has an inferred insert size smaller than expected. The mate-reads of green or gold colored reads were mapped to different chromosomes.

b. A representative read supporting the T-to-C artifact in Fig. 1a. The upstream sequence of the read (blue arrow) was mapped to the forward strand of the genome, and the downstream sequence of the same read (green arrow) was mapped to the reverse strand. **Strangely, the upstream and downstream sequences overlapped, as did the genomic sequences to which each was mapped. Since the upstream sequence was longer than the downstream sequence, only the upstream sequence was eventually mapped and the downstream sequence was soft-clipped.** Two palindromic sequences exist in close proximity to each other, and the mismatched base between the two sequences (red box) represent the source of the T-to-C artifact. Most of the downstream bases were soft-clipped.

c. **Presumed mechanism of the phenomenon observed in Fig. 1b.** Two palindromic sequences in a single-stranded DNA (ssDNA) formed a hairpin structure at the end-repair step of library preparation. After nicking and partial denaturation, the double-stranded DNA was regenerated during the end-repair step of library preparation. The mismatched base between two palindromic sequences was defined as a mutation.

c) Previous minor point #3: Please indicate the number of samples included in Fig. 3 (formerly Fig. 1e) and the type of tissue in the legend. Please also state why/how the 97 specific mutations were selected. Additionally, do the VAFs shown represent the mean or median of all samples analyzed? Some of this information is available from the point-by-point response but please make it clear for readers of the manuscript itself.

Thank you for pointing out what needs to be improved in Figure 3.

- The 97 specific mutations were derived from 31 FFPE normal breast tissue samples, 12 FFPE breast tumor samples, two fresh frozen normal breast tissue samples, and six fresh frozen breast tumor samples.
- The 97 mutations were randomly selected so that the ratio of true mutations to artifacts predicted by MicroSEC was approximately 1:2.
- Each shape in Fig. 3 shows the variant allele frequencies in amplicon-based sequencing and capture-based sequencing for a specific mutation detected in a sample. Thus, each VAF is a single value, not the mean or median.

The following sentences were **modified**.

Lines 226–229: The mutations were randomly selected from 31 FFPE normal breast tissue samples, 12 FFPE breast tumor samples, two fresh frozen normal breast tissue samples, and six fresh frozen breast tumor samples.

Lines 346–349: Ninety-seven mutations were randomly selected so that the ratio of true mutations to artifacts predicted by MicroSEC was approximately 1:2. The mutations were derived from 31 FFPE normal breast tissue samples, 12 FFPE breast tumor samples, two fresh frozen normal breast tissue samples, and six fresh frozen breast tumor samples.

Fig. 3 legend: The MicroSEC analysis results were validated with amplicon-based sequencing that enriches target genomic regions by PCR. Ninety-seven mutations were randomly selected from 31 FFPE normal breast tissue samples, 12 FFPE breast tumor samples, two fresh frozen normal breast tissue samples, and six fresh frozen breast tumor samples. Each shape shows the variant allele frequencies (VAFs) in amplicon-based sequencing and capture-based sequencing for a specific mutation detected in a sample. The mutations that passed through the MicroSEC filter were detected with a similar level of VAF by both capture-based sequencing and amplicon-based sequencing (blue), with the exception of a CENPA mutation. Of the five potential CG-to-TG artifacts (red), two mutations in ESR1 or BRCA2 were not amplified by amplicon-based sequencing. Filtered out mutations were not detected by amplicon-based sequencing (green), with the exception of a NFYA mutation.

d) Previous minor point #6: Thank you for adding the number of mutations to Supplementary Fig. 2. Please also state in the legend that the somatic mutations shown represent those present in normal breast tissue but not in normal blood. Please also state in the Results section the rationale for why normal breast tissue was compared to blood to identify somatic mutations. Based on the description scattered in various section of the manuscript, the authors first tested the algorithm in normal breast tissue so that they can test the efficacy (sensitivity) of the filtering. This was followed by a testing on specificity using 23 FF and 33 FFPE breast cancer samples (lines 154-160). An outline of study design can be of great help so that the readers can understand why the pipeline was first tested using normal breast tissues.

The following points have been added to the main text and Fig. S2.

- We noted that somatic mutations in each sample were determined by comparison with sequence data from normal blood sample from the same individuals.
- Not all mutations detected in the blood samples were germline mutations because some mutations could be caused by clonal hematopoiesis or sequencing errors. With this limitation clearly stated, mutations detected in blood samples were considered as germline mutations in this study.
- An outline of study design was described in the first section of Results, “Study design”.

The following sentences were modified.

Lines 139–146: We first tested the sensitivity of the algorithm using normal breast tissue, because there were few true mutations in normal mammary tissue and most mutations detected in FFPE samples were considered to be artifacts. This was followed by a testing on specificity using 23 FF and 33 FFPE breast cancer samples. After confirming the performance, MicroSEC was applied to the clinical sequencing and whole exome sequencing data to investigate the usefulness of MicroSEC in actual clinical practice. Since MICR-originating artifacts were not produced by PCR, amplicon-based sequencing was used as an external validation of MicroSEC.

Lines 147–155: We examined the **performance** of MicroSEC in distinguishing true mutations from FFPE artifacts with our custom-made multi-gene panel test, “Todai OncoPanel”(1). The panel including 15,600 capture probes were designed to examine 478 cancer-related genes. As the total size of target regions was 3.4 megabases, the average length of captured regions was approximately 220 base pairs. We obtained fresh frozen samples of normal blood from all cases. The somatic mutations were defined as those that were identified in sample DNA but absent from matched normal blood DNA, although not all mutations detected in the blood samples were germline mutations because some mutations could be caused by clonal hematopoiesis or sequencing errors.

Lines 157–159: To test the sensitivity of the filtering algorithm, we analyzed the target sequencing data of 53 FF and 190 FFPE normal breast tissue samples with a high mean coverage of 400 or more.

Lines 364–365: Somatic mutations in each sample were determined by comparison with sequence data from normal blood from the same individuals.

Fig. S2 legend: a. FFPE samples of normal breast tissues (n = 190) with total somatic mutations (upper), mutations filtered out by MicroSEC filter (middle), and mutations passing through the filter (lower). b. FFPE samples of breast tumor tissues (n = 33) with total somatic mutations (upper), mutations filtered out by MicroSEC filter (middle), and mutations passed through the filter (lower). The somatic mutations shown represent those present in normal breast tissue but not in normal blood. FFPE, formalin-fixed and paraffin-embedded.

e) Previous minor point #12: Thank you for the clarification of the FFPE-only variants in Supplementary Fig. 4. One minor typo may have been introduced: “fur to tumor heterogeneity”, where I believe the authors meant “due to tumor heterogeneity.”

The misspellings you pointed out have been corrected.

The following sentence was **modified**.

Line 189: The mutations found in each site were different **due to tumor heterogeneity**.

Additional new concerns

1. The previous manuscript lacked headers for “Introduction”, “Results”, etc. and these headings have been added to the current revised manuscript. However, the Introduction (e.g. lines 75-89; lines 115-131) actually contains Methods and Results material, including 2 of 3 main figures (Figs. 1 and 2) being cited and described in the Introduction. I would suggest moving all/most figure descriptions to the Results section, while the Introduction should retain background information and brief transitional material such as lines 90-98 and lines 109-115, which could help to nicely wrap up the Introduction and lead into the Results section. The algorithm design described in lines 115-131 can be put into a new Result section as “Study Design”. This section perhaps also should include description on method validation approaches using the three cohorts described in Table 1.

Thank you for the recommendation.

- We created sections “Examples of artifacts and the presumed mechanisms”, “Study design”, “Performance of MicroSEC”, “Hyperparameter optimization”, “Amplicon sequencing”, and “Application of MicroSEC to Clinical Sequencing and whole exome sequencing” in Results and reorganized the text as you described.

The following sentences were **modified**.

Lines 110–111: **The following phenomena are observed when reviewing all the reads supporting a mutation.**

Lines 139–155: **We first tested the sensitivity of the algorithm using normal breast tissue, because there were few true mutations in normal mammary tissue and most mutations detected in FFPE samples were considered to be artifacts. This was followed by a testing on specificity using 23 FF and 33 FFPE breast cancer samples. After confirming the performance, MicroSEC was applied to the clinical sequencing and whole exome sequencing data to investigate the usefulness of MicroSEC in actual clinical practice. Since MICR-originating artifacts were not produced by PCR, amplicon-based sequencing was used as an external validation of MicroSEC.**

We examined the **performance** of MicroSEC in distinguishing true mutations from FFPE artifacts with our custom-made multi-gene panel test, “Todai OncoPanel”. The panel including 15,600 capture probes were designed to examine 478 cancer-related genes. As the total size of target regions was 3.4 megabases, the average length of captured regions was approximately 220 base pairs. **We obtained fresh frozen samples of normal blood from all cases. The somatic mutations were defined as those that were identified in sample DNA but absent from matched normal blood DNA, although not all mutations detected in the blood samples were germline mutations because some mutations could be caused by clonal hematopoiesis or sequencing errors.**

2. Is Fig. 2a an actual read or hypothetical? From what type of sample? It would be helpful to explain the details.

Thank you for pointing this out.

- All reads in Fig. 2 are hypothetical prepared for the description of the method.

The following sentence was **added**.

Fig. 2 legend: **The principle of the algorithm is described with hypothetical reads.** a. Definition of supporting lengths. Supporting lengths are defined as the distances from the mutated base to the 5' or 3' ends of an individual read (excluding soft-clipped bases). The shorter supporting length is defined as the shorter one.

3. Lines 164-172 presented the following results “Sixty-five mutations were detected in FFPE samples only, of which 11 were detected as artifacts.”. The authors tried to justify that the FFPE-only variants were valid due to intratumor heterogeneity. However, the normal breast FFPE tissue filtering rate presented in Supplementary Figure 3 appears to have much higher number of mutations being filtered (5 per sample) even at the highest mutation depth bin of (90-100). It is important that the authors compare the filtering rate on the breast cancer samples with those presented in Supplementary Figure 3 to ensure that the low number of mutations being filtered in FFPE breast cancer tissue was indeed due to elevated coverage.

As you pointed out, the coverage of mutations detected in the FFPE samples was elevated since the number of subclones in the FFPE samples was lower than that in the fresh frozen samples. We have added this result as Fig. S4b and further described it in the text.

The following sentence was **added**.

Lines 187–195: DNA was extracted from frozen specimens of approximately 3 mm and thinly sliced FFPE specimens. The mutations found in each site were different **due** to tumor heterogeneity. Since frozen specimens consisted of multiple subclones, only common mutations were detected with VAF >5%, whereas FFPE specimens comprised only a small number of subclones, subclone-specific mutations were thus also detected. **The fact that mutations detected in FFPE samples had greater mutation coverage than those detected in FF samples supported this theory (Supplementary Fig. 4b).** This was the reason why 56 unfiltered mutations were detected in the FFPE samples, which we consider to be true mutations.

Fig. S4 was **modified**.

a Target sequencing on matched samples of breast cancer

c Whole exome sequencing on matched samples of primary cancer

b

Supplementary Figure 4. The mutations detected in matching FF and FFPE samples.

a. The somatic mutations detected in the target sequencing of matching sets of frozen (blue) and FFPE (peach) breast cancer samples from eight patients. Eleven mutations found in only FFPE samples were filtered by MicroSEC (green). All mutations found in fresh frozen samples passed through the MicroSEC filter. **b. Violin plots and box plots of mutation depths in matched FFPE and fresh frozen breast cancer samples. In target sequencing, mutations detected in FFPE samples showed higher mutation depths than those detected in frozen samples.** c. The somatic mutations detected in the whole exome sequencing of matching sets of frozen (blue) and FFPE (peach) primary cancer samples from 14 patients.

The 123 mutations (21.0%) found in only FFPE samples were filtered by MicroSEC (green). Eight mutations (0.5%) found in fresh frozen samples were filtered by the MicroSEC filter. FFPE, formalin-fixed and paraffin-embedded.

4. The exome FFPE analysis on lines 228-238 is very helpful; thank you for this addition. Could the authors also state the relative prevalence of CG-to-TG artifacts vs. MicroSEC-filtered (MICR-related) artifacts? This appears to be shown in Supp. Table 3 but a text description would also be helpful so that users can know approximately what percentage of their total artifact burden will be removed by MicroSEC.

Thank you for your comment, we have added a description of the frequency of CG-to-TG mutations and their ratio to MicroSEC-filtered artifacts.

The following sentences were **added**.

Lines 266–270: **CG-to-TG mutations were detected at a high rate of 38.7% of the total somatic mutations (45.8 mutations per sample), and thus most of these were possible artifacts. The ratio of CG-to-TG mutations to MicroSEC-filtered artifacts was 4.45:1, but this ratio might vary depending on the sample conditions and analysis methods.**

5. Line 151-152: “A tendency to increase could be observed for the mutation filtering rate as the number of reads with mutations increased (Supplementary Fig 3)”. This statement appears to be contradictory to the data presented in Supplementary Fig 3 and the S. Fig. 3 legend which states “The mutation filtering rate (black line) is high at low depth and more than 90% of mutations are filtered by MicroSEC pipeline when the mutation coverage is 10–22.”

We had stated the opposite. We have corrected the word "increase" to "decrease" as shown below.

The following sentence was **modified**.

Line 172: A tendency to **decrease** could be observed for the mutation filtering rate as the number of reads with mutations increased (Supplementary Fig. 3).

6. The validation described in lines 199-220: please clarify that these are the mutations detected in the normal breast tissue or the breast cancer samples.

Thank you for the comment. In addition to the response to the previous concern c), we have clearly indicated the samples in which the mutations were detected.

The following sentence was **added**.

Lines 226–229: The MicroSEC analysis results were validated with amplicon-based sequencing that enriches target genomic regions by PCR. Ninety-seven mutations including germline mutations and low VAF ones, found in the breast tissue samples, were examined. **The mutations were randomly selected from 31 FFPE normal breast tissue samples, 12 FFPE breast tumor samples, two fresh frozen normal breast tissue samples, and six fresh frozen breast tumor samples.**

7. Lines 239-266: the content should be restructured into Discussion.

Thanks for the suggestion. We have responded as such.

The following header was **added**.

Line 272: **Discussion**

REVIEWERS' COMMENTS:

Reviewer #1 (Remarks to the Author):

The authors have addressed all the questions that I raised.